# Dental morphology in *Homo habilis* and its implications for the evolution of early *Homo*

Thomas W. Davies [1,2] ✉, Philipp Gunz [1], Fred Spoor [1,3], Zeresenay Alemseged [4], Agness Gidna[5], Jean-Jacques Hublin[6,7], William H. Kimbel[8,13], Ottmar Kullmer [9,10], William P. Plummer[2], Clément Zanolli [11] & Matthew M. Skinner[2,7,12]

The phylogenetic position of *Homo habilis* is central to debates over the origin and early evolution of the genus *Homo*. A large portion of the species hypodigm consists of dental remains, but they have only been studied at the often worn enamel surface. We investigate the morphology of the *H. habilis* enamel-dentine junction (EDJ), which is preserved in cases of moderate tooth wear and known to carry a strong taxonomic signal. Geometric morphometrics is used to characterise dentine crown shape and size across the entire mandibular and maxillary tooth rows, compared with a broad comparative sample (*n* = 712). We find that EDJ morphology in *H. habilis* is for the most part remarkably primitive, supporting the hypothesis that the *H. habilis* hypodigm has more in common with *Australopithecus* than later *Homo*. Additionally, the chronologically younger specimen OH 16 displays a suite of derived features; its inclusion in *H. habilis* leads to excessive levels of variation.

The origins of the genus *Homo* remain elusive. Over 60 years ago, Leakey et al.[1] proposed the species *Homo habilis* with the discovery of fossils at Olduvai Gorge in Tanzania; they proposed that the species occupied a morphological gap between *Australopithecus* and *Homo erectus*, and placed it at the base of the genus *Homo*. While the naming of this new species was initially controversial, subsequent discoveries of early *Homo* specimens at Olduvai, as well as other sites in Kenya, South Africa, and Ethiopia, gradually led to the general acceptance of *H. habilis* as a valid species[2–7]. Despite this acceptance, a number of key questions about early *Homo* remain unresolved, and in particular, we have a poor understanding of the phylogenetic relationship between *H. habilis* and other species. Additionally, recent work has emphasised hominin species diversity in the Pliocene and Pleistocene, including the co-existence of multiple species in close proximity[8–12], and thus a

once influential account of hominin evolution, according to which *Australopithecus* evolved into *H. habilis*, then *H. erectus*, and ultimately modern humans[13,14], is now considered overly simplistic[15,16]. In the case of the early *Homo* fossil record; a number of studies have suggested that the variation exceeds that expected of a single species and that a second species, *Homo rudolfensis*, is also present[17–21]. Further support for the existence of this second species comes from a reconstruction of the mandible of the type specimen of *H. habilis*, OH 7, which found the specimen shows a primitive long and narrow dental arcade that is incompatible with several *H. rudolfensis* specimens[22]. Further, our understanding of the origins of the genus has been impacted by several discoveries of *Homo* fossils older than 2 million years ago (Ma)[7,23–28], including a mandible from Ledi Geraru, which represents the earliest known *Homo* at 2.8 Ma[29]. It is also worth noting that some

[1]Department of Human Origins, Max Planck Institute for Evolutionary Anthropology, Leipzig, Germany. [2]School of Anthropology and Conservation, University of Kent, Canterbury, UK. [3]Centre for Human Evolution Research, Natural History Museum, London, UK. [4]Department of Organismal Biology and Anatomy, University of Chicago, Chicago, IL, USA. [5]Department of Cultural Heritage, Ngorongoro Conservation Area Authority, P. O. Box 1Ngorongoro Crater, Arusha, Tanzania. [6]Collège de France, Paris, France. [7]Max Planck Institute for Evolutionary Anthropology, Leipzig, Germany. [8]Institute of Human Origins, and School of Human Evolution and Social Change, Arizona State University, Tempe, AZ, USA. [9]Palaeobiology and Environment workgroup, Institute of Ecology, Evolution, and Diversity, Goethe University, Frankfurt, Germany. [10]Division of Palaeoanthropology, Senckenberg Research Institute and Natural History Museum Frankfurt, Frankfurt am Main, Germany. [11]Univ. Bordeaux, CNRS, MCC, PACEA, UMR 5199, 33600 Pessac, France. [12]Centre for the Exploration of the Deep Human Journey, University of the Witwatersrand, Johannesburg, South Africa. [13]Deceased: William H. Kimbel. ✉e-mail: thomas_davies@eva.mpg.de

suggest that specimens currently assigned to *H. habilis* (particularly OH 13, OH 24, OH 62, and KNM-ER 1813) may in fact be better classified as *Australopithecus* due to a lack of synapomorphies with later *Homo*[30–32]. Others have suggested that the entire hypodigms of *H. habilis* and/or *H. rudolfensis* should be transferred out of the genus *Homo*[10,33–36], although several phylogenetic analyses support the monophyly of the genus as currently defined[37], with the inclusion of *Australopithecus sediba*[38,39], or with the exclusion of either *Homo floresiensis*[38] or *Homo naledi*[39].

The *H. habilis* hypodigm includes numerous dental remains, with OH 7 preserving a nearly complete mandibular tooth row, OH 13 and OH 16 preserving most mandibular and maxillary teeth, and OH 24 preserving most postcanine maxillary tooth positions. Several features of the dentition have been suggested to be derived relative to *Australopithecus*; the initial description of *H. habilis* highlighted relatively large anterior teeth and buccolingually narrow postcanine teeth[1], while subsequent work has highlighted the small size of the postcanine teeth, the reduced premolar talonids, and a reduced $M_3$ that is similar in size to the $M_2$[5]. However, the usefulness of a number of these traits in distinguishing *H. habilis* from *Australopithecus* has been questioned[22,40,41], and *H. habilis* teeth are considered by some to be relatively generalised in their morphology[19]. Importantly, these studies have only considered the outer-enamel surface (OES) morphology of these specimens, the original morphology of which is frequently altered or removed by the effects of occlusal dental wear. For example, the lower first molars of OH 7, OH 13, and OH 16 are moderately to heavily worn, and consequently, analyses of crown morphology have focused on 2D measurements of the occlusal surface (e.g., linear dimensions, relative cusp areas, and crown outline shape). Critically, these measurements do not capture the vertical components of both crown height and cusp height, which have been shown to be useful in distinguishing *Australopithecus* from *Homo*[42,43].

In this study, we overcome the limitations of variable dental wear and 2D measurements through an analysis of the dentine crown that underlies the enamel cap. Hereafter referred to as the enamel-dentine junction (EDJ), it is a surface that is established early in tooth development and preserves the form of the basement membrane upon which enamel is deposited. In addition to retaining the vertical components of crown and cusp height, it is where the majority of features that are known to carry taxonomic relevance are formed during odontogenesis. Since the EDJ is not remodelled throughout an individual's lifetime, it can provide insights into dental morphology that would otherwise not be possible due to the effects of tooth wear. A number of studies have demonstrated that the morphology of the EDJ is taxonomically informative in extant primates[42,44–46] and fossil hominoids[43,47–51], including a recent study in which Zanolli et al.[52] suggested that the *Homo* status of a number of South African fossils from Sterkfontein, Drimolen and Swartkrans is not supported based on the morphology of the EDJ. Fossils from Olduvai Gorge are crucial to the issues surrounding early *Homo* systematics, but little is known about their internal dental morphology.

Here we assess the EDJ morphology of key specimens from Olduvai (Figs. 1 and 2), including the type specimen of *H. habilis*, OH 7, as well as early *Homo* fossils from Koobi Fora. Using 3D landmark-based geometric morphometrics (GM), we characterise the major EDJ shape differences between *Australopithecus* and later *Homo* (modern humans, Neanderthals, and Middle-Pleistocene *Homo*; Table 1). We find that key *H. habilis* specimens including OH 7 retain the primitive condition and that several features previously suggested to characterise the dentition of *H. habilis* are not evident at the EDJ. We also assess the homogeneity of the *H. habilis* hypodigm from a dental perspective, finding high levels of variation that in some cases exceed our expectations of a single species.

## Results

### Overall shape patterns

Principal component analyses (PCA) summarise EDJ shape variation for three representative tooth positions; $P_3$ (Fig. 3a), $P^4$ (Fig. 4a), and $M^1$ (Fig. 5a). PCAs for all postcanine tooth positions can be found in

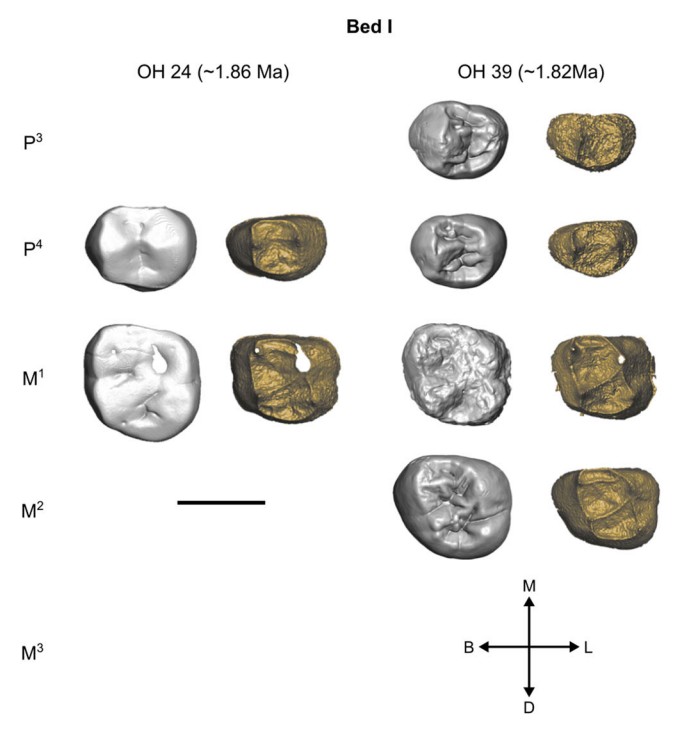
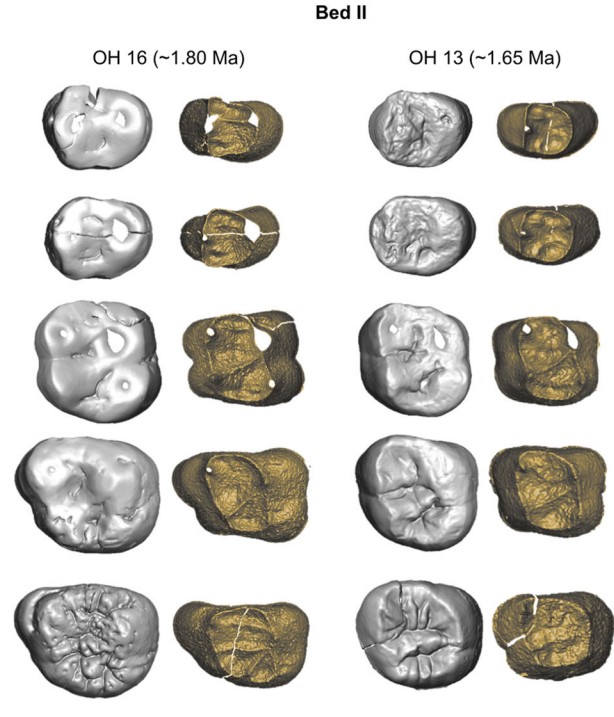

**Fig. 1 | Olduvai *Homo habilis* maxillary tooth rows.** Three *H. habilis* tooth rows are shown, OH 7, OH 13, OH 16, OH 39. Each is shown in the occlusal view at the outer-enamel surface (left) and enamel-dentine junction (right). All teeth are shown as right-sided. Dating and stratigraphy information from refs. 69,72,73.

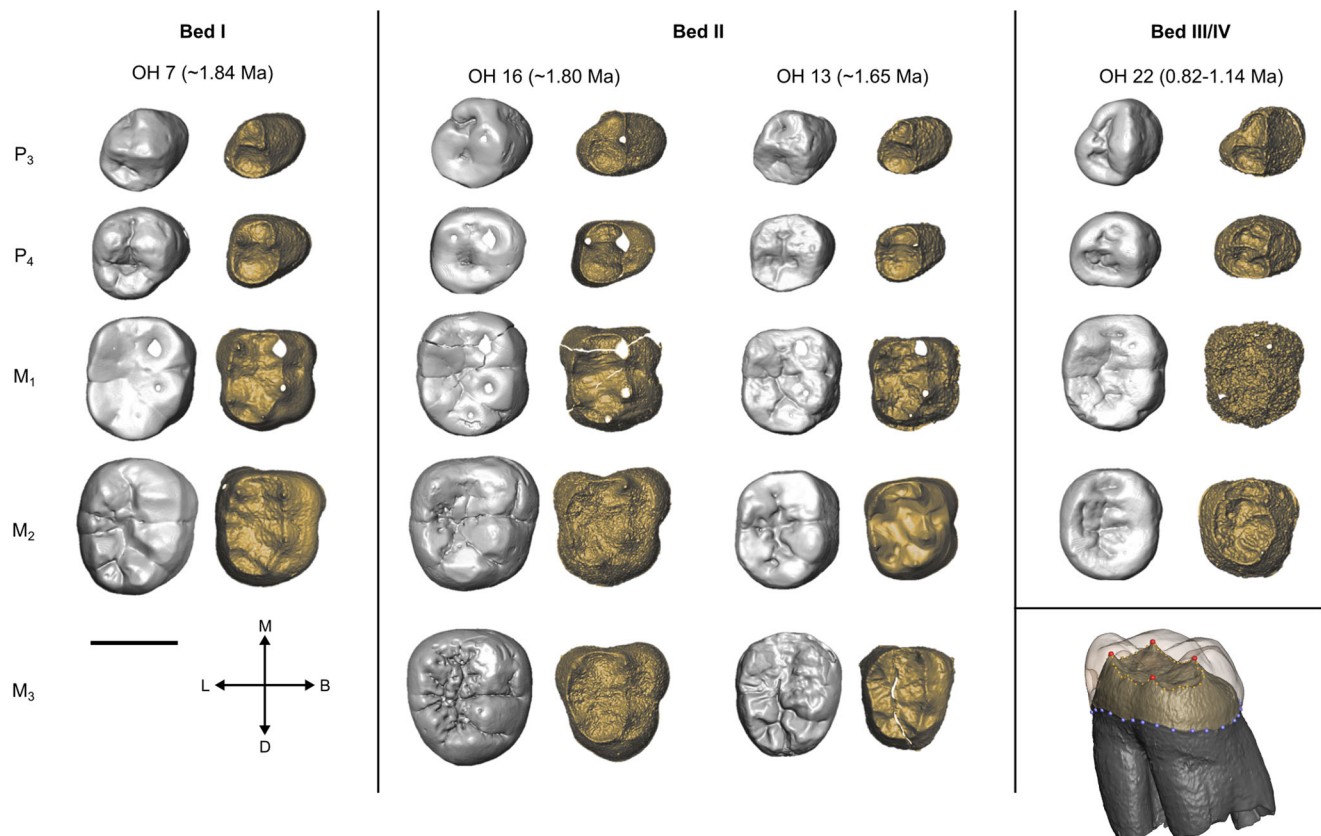

**Fig. 2 | Olduvai mandibular tooth rows.** Three *H. habilis* tooth rows are shown, OH 7, OH 13, and OH 16, as well as OH 22 which is attributed to *H. erectus*. Each is shown in the occlusal view at the outer-enamel surface (left) and enamel-dentine junction (right). All teeth are shown as right-sided. An example of the landmarking protocol for molars is shown in the bottom right (Red = EDJ fixed landmarks, yellow = EDJ ridge semilandmarks, blue = CEJ ridge semilandmarks). Dating and stratigraphy information from refs. 69,72,73,87,88.

## Table 1 | Sample summary

| Taxon | Locality | I₁ | I₂ | C | P₃ | P₄ | M₁ | M₂ | M₃ |
|---|---|---|---|---|---|---|---|---|---|
| *Australopithecus* | Hadar, Ethiopia; Laetoli, Tanzania; Sterkfontein and Makapansgat, South Africa | 6 | 7 | 10 | 12 | 16 | 15 | 22 | 16 |
| *Homo habilis* | Olduvai Gorge, Tanzania; Koobi Fora, Kenya | 2 | 2 | 3 | 4 | 5 | 6 | 5 | 4 |
| Ungrouped | Koobi Fora, Kenya; Omo, Ethiopia | - | - | - | 2 | 1 | 2 | 1 | - |
| *Homo erectus* | Koobi Fora, Kenya; Olduvai Gorge, Tanzania; Sangiran, Indonesia | 2 | 2 | 4 | 6 | 8 | 8 | 11 | 4 |
| Later *Homo* | Various Sites—see Supplementary Data 21 | 8 | 8 | 8 | 31 | 20 | 30 | 37 | 25 |
| **Taxon** | **Locality** | **I¹** | **I²** | **C** | **P³** | **P⁴** | **M¹** | **M²** | **M³** |
| *Australopithecus* | Hadar, Ethiopia; Laetoli, Tanzania; Sterkfontein and Makapansgat, South Africa | 9 | 9 | 8 | 13 | 13 | 18 | 16 | 12 |
| *Homo habilis* | Olduvai Gorge, Tanzania; Koobi Fora, Kenya | 2 | 1 | 4 | 4 | 5 | 6 | 4 | 4 |
| Ungrouped | Koobi Fora, Kenya; Hadar and Omo Ethiopia; Swartkrans, South Africa | - | 1 | 1 | 2 | 2 | 3 | 2 | 2 |
| *Homo erectus* | Koobi Fora, Kenya; Sangiran, Indonesia | 1 | 2 | 1 | 5 | 5 | 5 | 6 | 2 |
| Later *Homo* | Various Sites—see Supplementary Data 21 | 13 | 16 | 15 | 29 | 30 | 28 | 40 | 21 |

The hominin taxa included in the sample are listed, along with their locality and the sample size for each mandibular (top) and maxillary (bottom) tooth position.

Supplementary Fig. 7 (interactive html versions of these plots are available in Supplementary Data Files 1–10). In each case, the first principal component (PC1) separates *Australopithecus* (*Australopithecus afarensis* and *Australopithecus africanus*) from the later *Homo* sample. For most tooth positions, and particularly the pre-molars, PC1 represents a high proportion of the total shape variation present in the sample. This corresponds to a number of shape differences present throughout the postcanine dentition of the later *Homo* group that distinguish them from *Australopithecus*. These shape changes are summarised in Table 2 (see also Supplementary Tables 4 and 5) and visualised using wireframes (Figs. 3b, 4b, 5b; Supplementary Figs. 8 and 9). One notable shape difference between *Australopithecus* and later *Homo* is a relative increase in the height of the dentine body (see Supplementary Fig. 5 for an explanation of the terminology used to describe crown height at the EDJ) in the latter group; this is present across all postcanine tooth positions, although it is less pronounced in the upper molars. Further shape differences in later *Homo*, when compared with *Australopithecus*, include a

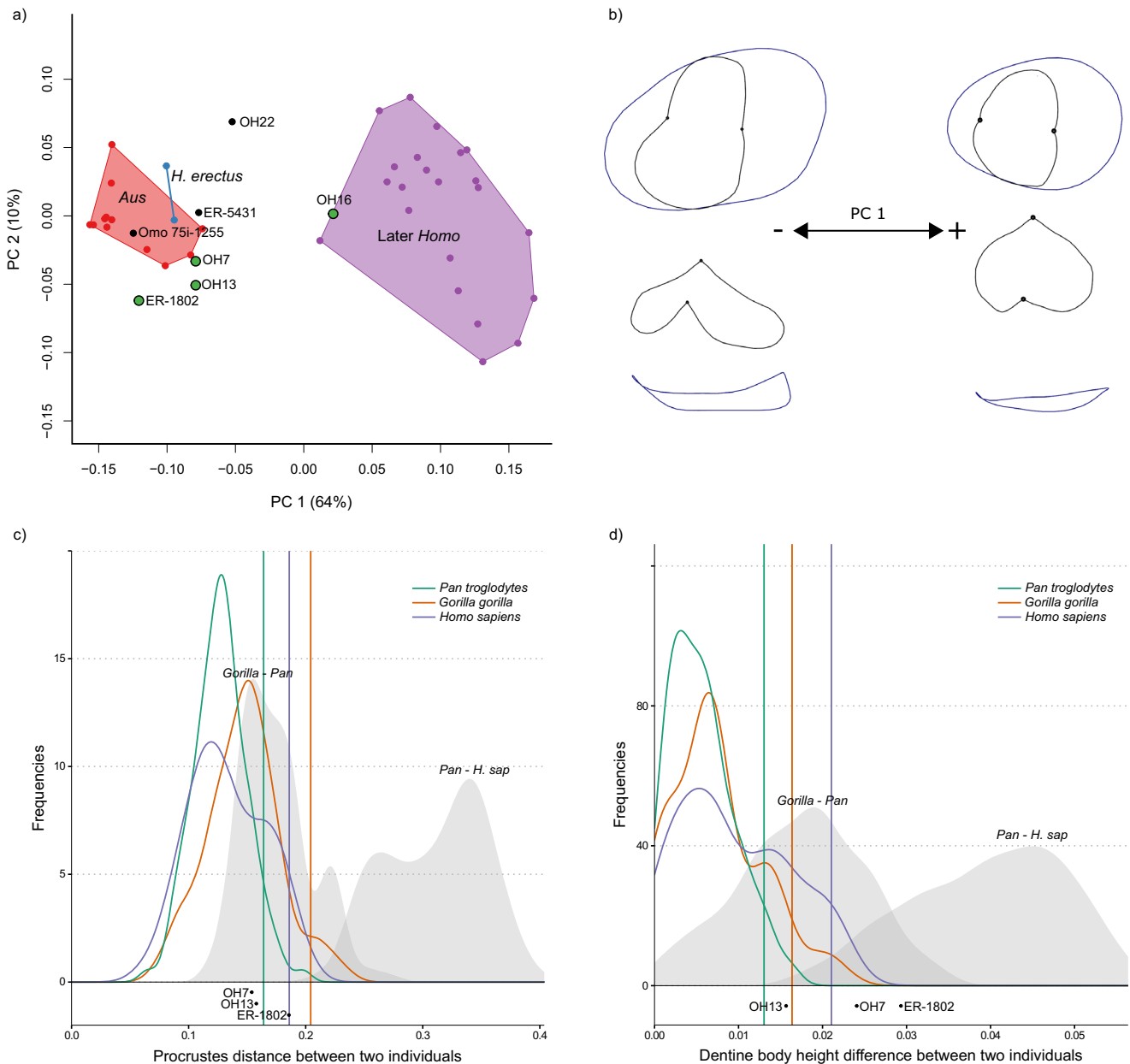

**Fig. 3 | EDJ shape variation in mandibular third premolars. a** PCA plot of P₃ EDJ shape. *Homo habilis* specimens are represented by green points. *Aus* = *Australopithecus*. PC = principal component. **b** Wireframe images showing shape changes associated with the first principal component, showing landmark positions associated with extremes of PC1 (±1.5 standard deviations from the mean) in occlusal view (top) and lingual view (bottom). **c** Frequency plot of Procrustes distances between all possible pairs of individuals within (coloured lines) and between groups (grey fill). Vertical lines show the 95% limits of the within-group distributions. Below is shown the Procrustes distance between OH 16 and other *H. habilis* or early *Homo* specimens. Red = *Gorilla gorilla*, Green = *Pan troglodytes*, Purple = *Homo sapiens*. *H. sap* = *H. sapiens*. **d** Same as **c** for dentine body height. Source data are provided as a Source Data file.

reduction in the talon/talonid region in the premolars, a relatively taller and more distally placed protoconid in the lower molars, and a reduction in the distal marginal ridge in the upper molars (Table 2). The difference in EDJ shape between later *Homo* and *Australopithecus* is significant for all tooth positions in permutation tests (Supplementary Table 6).

In these features that serve to distinguish between *Australopithecus* and later *Homo* at the EDJ, a number of *H. habilis* specimens retain the primitive condition, including the holotype, OH 7. For example, the mandibular premolar talonids are not reduced and the relative dentine body height is either short (M₂) or intermediate (P₃, P₄, M₁). Table 2 summarises the condition observed in selected Olduvai *H.*

*habilis* specimens for these traits, and Supplementary Table 4 for specimens from Koobi Fora and A.L. 666-1. Other Bed I Olduvai specimens OH 4, OH 24, and OH 39 also approximate the *Australopithecus* condition, as do Koobi Fora *H. habilis* specimens KNM-ER 1502, KNM-ER 1802 and KNM-ER 1813 from Koobi Fora. This is evident from the GM analysis, in which these specimens plot closely to specimens of *Australopithecus*. Furthermore, we found only a few significant differences in postcanine tooth shape or size between *Australopithecus* and *H. habilis* in permutation tests (Supplementary Table 6). Several tooth positions of KNM-ER 5431 (Hominini gen. et sp. indet.) and A.L. 666-1 from Hadar (*Homo* aff. *H. habilis*) also show similar EDJ morphologies to *Australopithecus*.

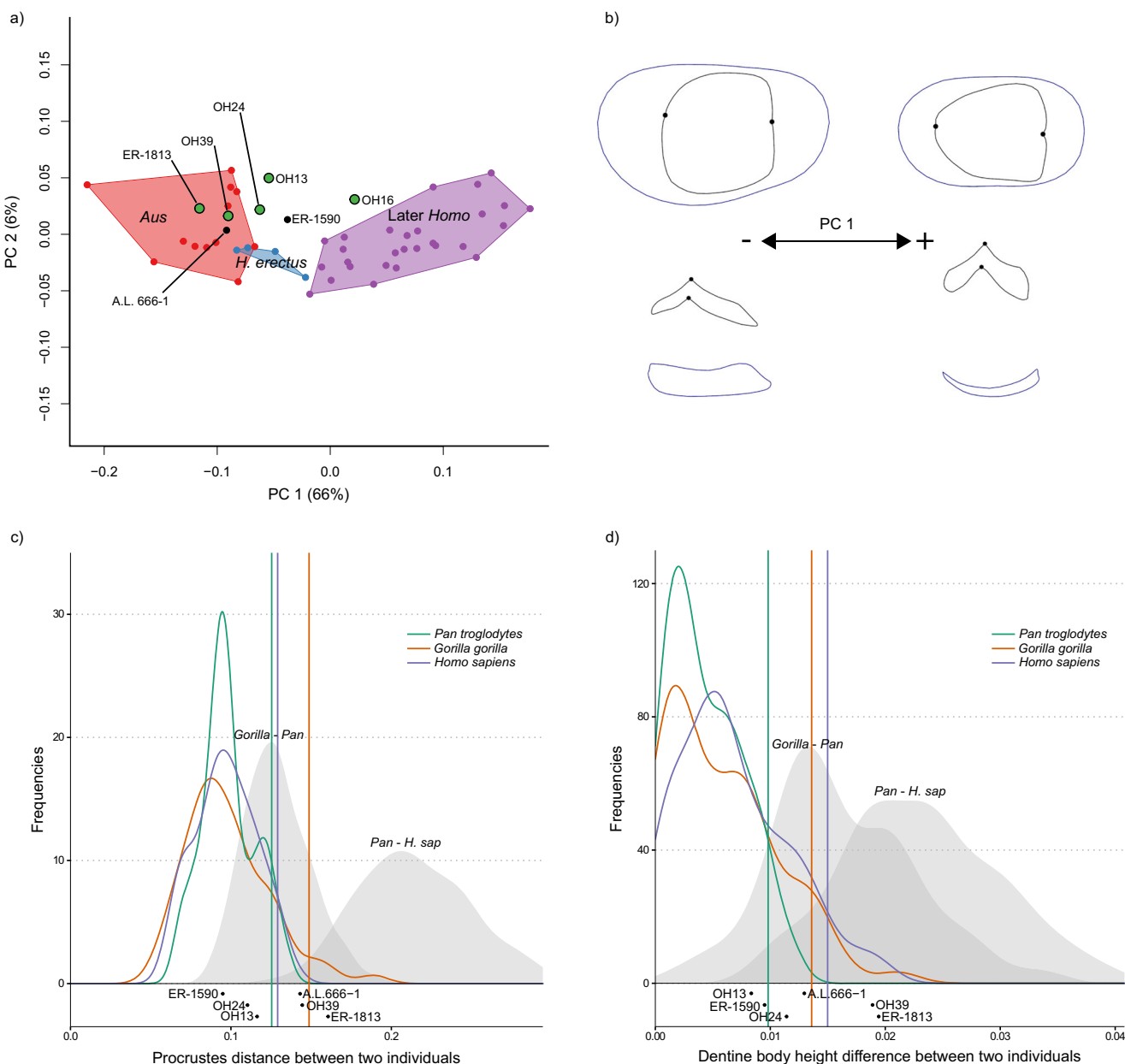

**Fig. 4 | EDJ shape variation in maxillary fourth premolars. a** PCA plot of P⁴ EDJ shape. *Homo habilis* specimens are represented by green points. *Aus* = *Australopithecus*. PC = principal component. **b** Wireframe images showing shape changes associated with the first principal component, showing landmark positions associated with extremes of PC1 (±1.5 standard deviations from the mean) in occlusal view (top) and lingual view (bottom). **c** Frequency plot of Procrustes distances between all possible pairs of individuals within (coloured lines) and between groups (grey fill). Vertical lines show the 95% limits of the within-group distributions. Below is shown the Procrustes distances between OH 16 and other *H. habilis* or early *Homo* specimens. Red = *Gorilla gorilla*, Green = *Pan troglodytes*, Purple = *Homo sapiens*. *H. sap* = *H. sapiens*. **d** Same as **c** for dentine body height. Source data are provided as a Source Data file.

Among specimens commonly attributed to *H. habilis*, the most derived condition is seen in OH 16. This is particularly evident in the premolars, which show a tall relative dentine body height, reduced talons/talonids ($P_3$, $P_4$), and an interrupted $P_3$ mesial marginal ridge, features that are frequent in later *Homo*. The molars also show a tall dentine body, as well as taller molar dentine horns, particularly the mandibular molar protoconids and maxillary molar protocones ($M^2$ and $M^3$; Table 2). These features contribute to the position of OH 16 in PCAs, which for some tooth positions is closer to later *Homo* than *Australopithecus*. However, it is important to note that there are also a number of primitive aspects of the EDJ morphology in OH 16, such as the shape of the EDJ marginal ridges in the $P_4$, $M_1$, $M^1$, and $P^3$, which are similar to that seen in other *H. habilis* specimens and *Australopithecus*.

OH 13 shows a mixed pattern; the $P_3$ and $M_2$ are most similar to OH 7 and KNM-ER 1802, while the $M^2$ and $M^3$ appear to be more derived, similar to OH 16.

The overall shape differences (measured using pairwise Procrustes distances; Figs. 3c, 4c, 5c; Supplementary Fig. 10) between OH 16 and other *H. habilis* specimens are mostly within the 95% limit of extant groups, although $P_4$ shape is distinct from KNM-ER 1802, the shape of the $M^1$ is distinct from OH 21, and the $P^4$ is distinct from KNM-ER 1813. However, the clearest differences between OH 16 and other *H. habilis* specimens are in relative dentine body height. When this feature is considered in isolation, OH 16 is distinct from OH 7 ($P_3$-$P_4$), OH 21 ($M^1$), OH 39 ($P^3$-$P^4$), KNM-ER 1802 ($P_3$-$P_4$), KNM-ER 1813 ($P^4$-$M^2$) and KNM-ER 1502 ($M_1$). In

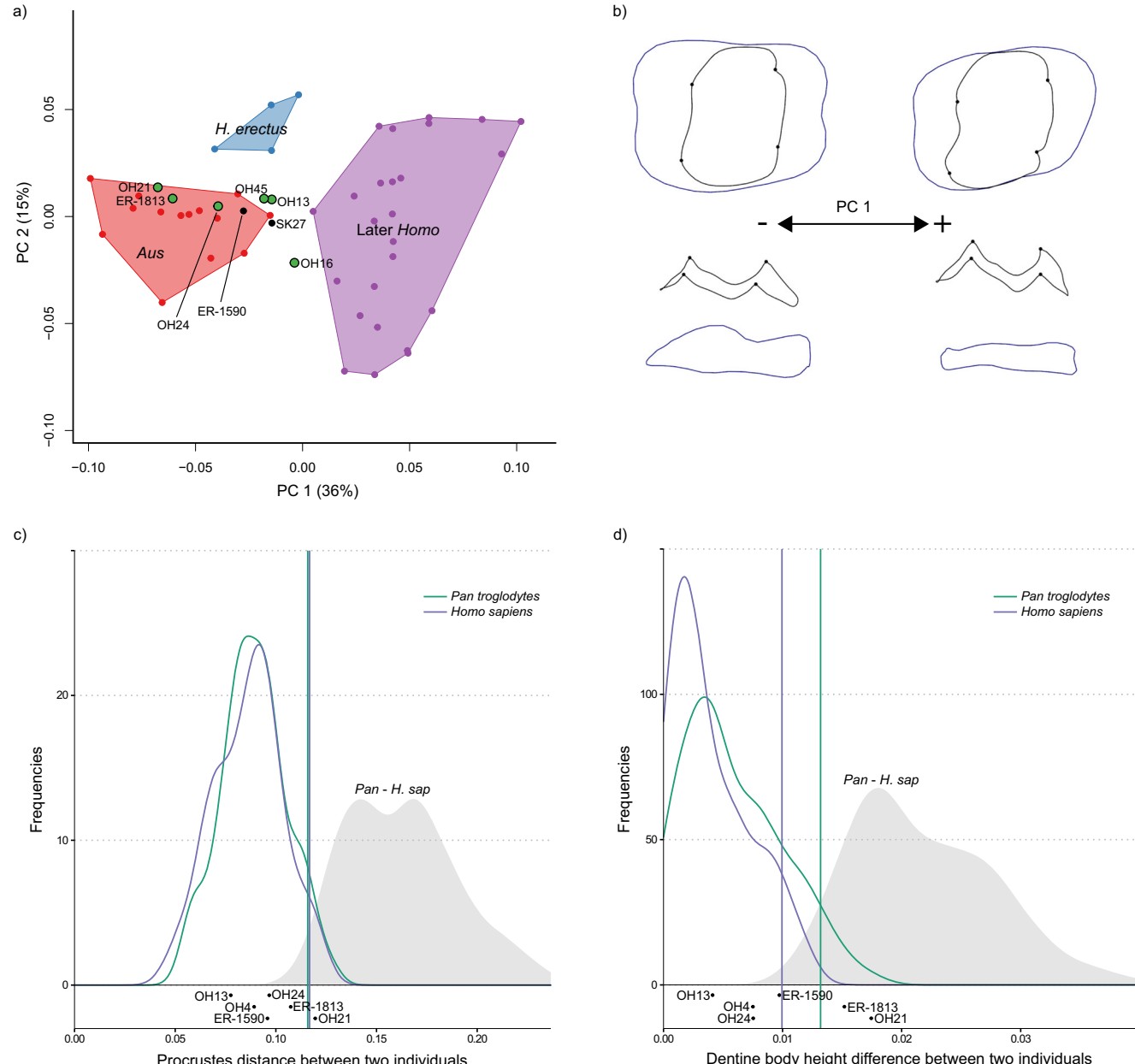

**Fig. 5 | EDJ shape variation in maxillary first molars. a** PCA plot of M$^1$ EDJ shape. *Homo habilis* specimens are represented by green points. *Aus* = *Australopithecus*. PC = principal component. **b** Wireframe images showing shape changes associated with the first principal component, showing landmark positions associated with extremes of PC1 (±1.5 standard deviations from the mean) in occlusal view (top) and lingual view (bottom). **c** Frequency plot of Procrustes distances between all possible pairs of individuals within (coloured lines) and between groups (grey fill). Vertical lines show the 95% limits of the within-group distributions. Below is shown the Procrustes distances between OH 16 and other *H. habilis* or early *Homo* specimens. Green = *Pan troglodytes*, Purple = *Homo sapiens*. *H. sap* = *H. sapiens*. **d** Same as **c** for dentine body height. Source data are provided as a Source Data file.

these cases, the difference in relative dentine body height between OH 16 and other specimens attributed to *H. habilis* exceeds that which would be expected of a single species (Figs. 3d, 4d, 5d; Supplementary Fig. 11).

Some tooth positions provide better distinction between specimens of *Australopithecus* and early *Homo* than others in PCA plots. For example, while the M$_2$s of *H. habilis* and *H. erectus* largely overlap with those of *Australopithecus*, there is a better distinction in other tooth positions, particularly the M$^3$ (Supplementary Fig. 6) despite their generally more variable shape. This is well illustrated in the example of KNM-ER 1813. The P$^4$, M$^1$, and M$^2$ of this specimen plot with *Australopithecus*, while the M$^3$ is clearly distinct. In this case, *H. habilis* specimens occupy the positive end of PC3, which corresponds to a

mesiodistally shortened crown, with an asymmetrical EDJ ridge due to particularly reduced mesiolingual and distobuccal crown corners. The M$^2$ also provides a better distinction between *H. habilis* and *Australopithecus* in most cases; only KNM-ER 1813 plots within the *Australopithecus* range. This is also reflected in permutation tests, which find that the mean shapes of the M$^2$ and M$^3$ differ significantly between *Australopithecus* and *H. habilis*. There was also a significant mean difference between these two groups in P$_3$ shape; this is driven partly by the derived condition seen in OH 16, but other *H. habilis* P$_3$s also have a slightly taller dentine body height than in *Australopithecus*, as well as a more symmetrical crown base, and an internally placed protoconid dentine horn tip (this feature could not be assessed in OH 16 due to wear).

**Table 2 | EDJ shape differences between *Australopithecus* and later *Homo*, and their condition in Olduvai specimens**

| Shape characteristics in later *Homo* | OH 7 | OH 13 | OH 16 | OH 24 |
|---|---|---|---|---|
| Increased relative dentine body height (all teeth) | Dentine body short ($M_2$) or intermediate ($P_3$, $P_4$, $M_1$) | Dentine body short ($M_2$), intermediate ($P_3$, $P_4$, $M_3$) or tall ($M_1$, $P^4$, $M^1$, $M^2$, $M^3$) | Dentine body tall (all teeth) | Dentine body intermediate ($P^4$) or tall ($M^1$) |
| Talon/talonid reduction ($P_3$, $P_4$, $P^3$, $P^4$) | No talonid reduction ($P_3$ and $P_4$) | No talonid reduction in $P_3$, $P^3$ slight/moderate talon/talonid reduction in $P_4$, $P^4$ | Talon/talonid clearly reduced ($P_3$,$P^4$) or slightly reduced ($P_4$,$P^3$) | Squared distolingual crown corner, no talon reduction ($P^4$) |
| Metaconid reduced relative to protoconid ($P_3$, $P_4$) | No metaconid reduction ($P_3$, $P_4$) | Metaconid reduced ($P_3$) or slightly reduced ($P_4$) | Metaconid reduced in $P_3$, not in $P_4$ | - |
| Protocone relatively taller and more distally placed ($M^1$, $M^2$, $M^3$)[a] | - | Intermediate protocone placement, but height as in *Australopithecus* ($M^1$, $M^2$, $M^3$) | Protocone taller and distally placed in $M^3$, taller and intermediately placed ($M^2$) or as in *Australopithecus* ($M^1$) | Protocone placement intermediate but taller ($M^1$) |
| Reduced distal marginal ridge (DMR) ($M^1$, $M^2$, $M^3$) | - | No DMR reduction in $M^1$, moderate reduction in $M^2$, clear reduction in $M^3$ | No DMR reduction in $M^1$, but clear distal reduction in $M^2$ and $M^3$ | No DMR reduction ($M^1$) |
| Protoconid are relatively taller and more distally placed ($M_1$, $M_2$, $M_3$)[a] | Protoconid moderately taller and more distally placed ($M_1$, $M_2$) | Protoconid not distally placed, and variable in height ($M_1$ moderately tall, $M_2$ tall, $M_3$ not taller) | Protoconid tall but not distally placed ($M_1$, $M_2$, $M_3$) | - |
| Rounded crown—EDJ ridge mesiodistally shorter and buccolingually wider ($M_1$, $M_2$, $M_3$)[b] | Less rounded crown, mesiodistally elongated ($M_1$, $M_2$) | $M_1$ marginally more rounded, $M_2$ similar to *Australopithecus*, $M_3$ mesiodistally longer | $M_1$ mesiodistally elongated, $M_2$ and $M_3$ more rounded | - |

The features listed are those identified through GM analyses that distinguish between *Australopithecus* and later *Homo*. The condition in later *Homo* is described in the first column and the condition in the Olduvai *H. habilis* specimens in the subsequent columns. A dash indicates that the specimen does not preserve the relevant tooth positions.
[a]First molars of *H. habilis* show wear on protocone/protoconid, these results are based on dentine horn reconstructions.
[b]*H. habilis* has mesiodistally elongated mandibular molars at the OES relative to *Australopithecus*; for more discussion of this trait, see Supplementary Note 1.

Canonical variate analysis (CVA) was carried out to further investigate shape differences between *Australopithecus*, *H. habilis*, and *H. erectus* (later *Homo* was excluded in this case in order to provide the best distinction between early *Homo* and *Australopithecus*). CVA results are shown in Supplementary Figs. 12 and 13, and cross-validated CVAs (cvCVA) in Supplementary Fig. 14. CVA plots show the distinction between *Australopithecus* and *H. habilis* in several tooth positions, however, in the majority of cases, separation between these groups is severely reduced or removed in cvCVAs, suggesting they likely represent spurious group differences. The differences between *H. habilis* and *Australopithecus* in the upper molars are maintained in cvCVAs however, indicating these represent real shape differences. In the $M^3$, the *H. habilis* specimens are distinguished from *Australopithecus* along CV2, which represents several of the same features described above (a mesiodistally short crown, asymmetrical at the EDJ ridge). The $M^2$s are similarly reduced mesiodistally, with an asymmetrical EDJ ridge due to a reduced mesiolingual and distobuccal crown corner, while the $M^1$s are distinguished along a combination of CV1 and CV2, corresponding to subtle differences in EDJ ridge shape around the metacone, crown height on the lingual side of the crown, and the shape of the cementum-enamel junction (CEJ; although it is worth noting that the shape difference between *Australopithecus* and *H. habilis* $M^1$s were not significant in permutation tests).

Several *H. erectus* specimens included here also plot more closely to *Australopithecus* than later *Homo* in PCAs, although this varies by specimen and tooth position, and specimens from Sangiran are better distinguished from *Australopithecus* than those from Koobi Fora. KNM-ER 1507 is the least derived; the specimen plots within the *Australopithecus* range along the first two PCs for $P_3$-$M_2$, and shows no talonid reduction in the premolars, a short or intermediate dentine body height, and a mesiodistally elongated crown in $M_1$ and $M_2$ (Supplementary Table 5). These features are more similar to *Australopithecus* or *H. habilis* than to other *H. erectus*, particularly specimens from Sangiran. KNM-ER 992 is more derived, and although several tooth positions overlap with *Australopithecus* along PC1, there is a moderately tall dentine body in each tooth, there is some reduction of the premolar talonids, and the $M_2$ and $M_3$ show more rounded EDJ marginal ridges. Other Koobi Fora *H. erectus* specimens plot either on the periphery or just outside the *Australopithecus* range in the first 2 PCs (KNM-ER 3733 $P^3$, $P^4$, KNM-ER 806 $M_1$) or are distinguished along PC2 (KNM-ER 806 $M_2$, $M_3$, KNM-ER 807 $M^1$, KNM-ER 1808 $M_2$) or PC3 (KNM-ER 807 $M^3$, KNM-ER 1808 $M^2$; Supplementary Fig. 15). OH 22 is well-distinguished from other specimens within the sample in PCAs, including those attributed to *H. erectus*. In particular, the specimen occupies a positive position along PC2 for $P_3$-$M_1$, and a positive position along PC3 for $M_2$ (Supplementary Fig. 15). This reflects a number of features across the tooth row, including the presence of a low mesial marginal ridge in both the premolars and molars, a relatively elongated mesial fovea (present in all teeth, but particularly notable in the $P_3$) and a short metaconid. *H. erectus* specimens are clearly distinct from both *Australopithecus* and *H. habilis* in CVAs, and these group differences are maintained in cvCVAs for most tooth positions, except for the $P^3$ where there is overlap between all three groups and the $M^2$ where KNM-ER 1808 and Sangiran 4 overlap with *Australopithecus*. This distinction corresponds to a number of features across the dentition including premolars with a more symmetrical, oval-shaped CEJ (crown base), more rounded EDJ marginal ridges molars, and a reduced maxillary molar hypocone.

Typicality-based classifications for unclassified specimens are presented in Supplementary Table 7. KNM-ER 5431 is mostly classified as *Australopithecus*. The premolars of KNM-ER 1590 and A.L. 666-1 are also classified as *Australopithecus*, however, the molars are frequently not classified into any of the groups. KNM-ER 1507 is classified mostly as *H. habilis* or *Australopithecus*. L26-1g, L398-573, and SK 847 are most often not classified into any group, while SK 27 is most often classified

as *Homo erectus*, and Omo 75i-1255 is most often classified as *H. habilis*. OH 22 plots either close to *H. erectus*, or occupies a more extreme position along the CV that best separates *H. erectus* from other taxa. In particular, the $P_4$ has an oval-shaped CEJ that is mesiodistally compressed and a clearly reduced talonid, while the $M^2$ has a rounded EDJ ridge, which is similar to other *H. erectus* specimens but more extreme in their expression. As a result of these features, the teeth are either classified as *H. erectus* or none of the groups.

Variation in CEJ shape in each anterior tooth position is summarised in Supplementary Fig. 16. The mandibular incisors provide a good distinction between *Australopithecus* and later *Homo* along PC1; this refers to the extent of the apical extension of the enamel on the labial and lingual sides of the tooth, which is more marked in *Australopithecus*, giving the CEJ a sinusoidal shape. In this respect, OH 7 and OH 16 retain the *Australopithecus* condition, while two *H. erectus* specimens (KNM-ER 820 and KNM-WT 15000) show a much flatter CEJ, even more so than in later *Homo* specimens. This CEJ shape difference is less marked in the other anterior tooth positions, and as a result, there is more overlap between groups in these cases.

### Buccolingually narrow postcanines

At the enamel surface, most *H. habilis* postcanine teeth are buccolingually narrower than those of *Australopithecus*, particularly in the mandibular dentition (Supplementary Tables 1 and 2). However, this pattern is less evident at the EDJ, instead, there is variation by specimen, tooth position, and by which part of the EDJ is being considered (Supplementary Figs. 1 and 2). This trait is discussed in detail in Supplementary Note 1.

### Size patterns across tooth row

Figure 6 shows the cervical size patterns across the tooth row in *H. habilis* when compared with *Australopithecus, H. erectus*, and later *Homo* (comparisons between *H. erectus* and other groups, including *Homo* sp., are available in Supplementary Fig. 17, interactive html versions of each centroid size plot are available in Supplementary Data files 11-20 and all centroid size data is available in Supplementary Data 21). The size of the mandibular teeth of *H. habilis* show a large degree of overlap with those of *Australopithecus*, and we found no significant size differences between the two taxa (Supplementary Table 6). In OH 7, all preserved tooth positions are within the range of *Australopithecus*. A possible exception to this pattern is OH 13, which is the smallest *H. habilis* specimen included here, and for which the canine and $P_4$ are outside the *Australopithecus* range of variation. OH 13 also differs from other *H. habilis* specimens in having a larger $P_3$ than canine; in OH 7 and OH 16 the $P_3$ is smaller. The $P_3 > C$ pattern seen in OH 13 is similar to that seen in a number of *H. erectus* specimens (KNM-ER 820, 992, 1507, and KNM-WT 15000). There is also an overlap in maxillary tooth size between *H. habilis* and *Australopithecus*, although the postcanine teeth occupy the lower end of the *Australopithecus* range, and some $M^2$s and $M^3$s are smaller. The teeth of OH 16 are large, similar to other *H. habilis* and *Australopithecus*. The anterior teeth in *H. habilis* are variable; OH 15 (C) and OH 16 ($I^1$) occupy the larger end or are slightly above, the *Australopithecus* range of variation, while KNM-ER 1813 ($I^2$-C) and OH 39 ($I^1$, C) occupy the lower end or are slightly below the *Australopithecus* range. *H. habilis* specimens mostly have maxillary premolars that are similar in size (or in OH 39, a $P^4$ that is slightly smaller), whereas *Australopithecus* usually has a $P^4 > P^3$ pattern. However, there are also some *A. afarensis* specimens in which the $P^3$ is larger (A.L. 199-1 and A.L. 200-1a).

The mandibular teeth of *H. habilis* are generally larger than those of later *Homo*, although the difference is only significant in the posterior tooth positions for which there are larger sample sizes (Supplementary Table 6). The maxillary teeth of *H. habilis* show more overlap with those of later *Homo*, but the premolars and molars were still found to be significantly different. There were no significant

differences between *H. habilis* and *H. erectus*, although a number of Sangiran *H. erectus* specimens and KNM-ER 820 ($P_4$-$M_2$) are below the *H. habilis* range. Most postcanine tooth positions in *H. erectus* were found to be significantly different to later *Homo* (except $M^1$ and $M^3$), and only a small number of tooth positions ($M^3$, $M_2$, and $M_3$) were significantly different to *Australopithecus*. The size of the teeth of OH 22 and OH 23 is similar to other *H. erectus* specimens (Supplementary Fig. 17), except that both show a $M_1 > M_2$ pattern, whereas KNM-ER 806, KNM-ER 992, KNM-ER 1507, KNM-WT 15000 and Sangiran 1b show the opposite pattern. The teeth of KNM-ER 1590 are large compared to those of *H. habilis*, and it has a canine that is similar in cervix size to the $P^3$, which is rare in *Australopithecus*, and not present in either of the *H. habilis* specimens that preserve these teeth (OH39 and KNM-ER 1813), or KNM-WT 15000, but is seen in some later *Homo*. In A.L. 666-1, the premolars are slightly larger than *H. habilis*, but the $M^1$ and $M^2$ are within the *H. habilis* range.

## Discussion

A suite of EDJ features distinguishes the postcanine teeth of later *Homo* from those of *Australopithecus* (*A. afarensis* and *A. africanus*), including the relative height of the dentine body and reduction of the premolar talons and talonids. However, when considering these traits in early *Homo*, we find that key *H. habilis* specimens such as OH 7, OH 24, and KNM-ER 1813 largely retain the *Australopithecus* condition. The EDJ shape and size patterns across the tooth row in these *H. habilis* specimens accord more closely with *Australopithecus* than later *Homo*, and we fail to find statistically significant differences between our *H. habilis* and *Australopithecus* samples in size in any tooth position, and only find significant shape differences in three tooth positions ($P_3$, $M^2$, $M^3$) (Supplementary Table 6).

In fact, we find the *H. habilis* $M^2$ and $M^3$ shapes to be distinct from those of other groups more broadly. *Homo habilis* $M^2$s ($n = 4$), and especially $M^3$s ($n = 4$), have a mesiodistally short, asymmetrical EDJ ridge. In the $M^2$s, this is due to a reduced crown distobuccally, while the $M^3$s are reduced mesiolingually and distobuccally. The $M^3$s also frequently show a metacone dentine horn that is mesially and internally placed, but this is less pronounced in the $M^2$s. This combination of features drives the separation of *H. habilis* in CVAs for both tooth positions, and the separation of $M^3$s along PC3 of the PCA (Supplementary Fig. 7), and distinguishes them from *Australopithecus*, later *Homo*, and *H. erectus* (*H. erectus* $M^3$s also have an EDJ ridge that is shortened mesiodistally, but they are less asymmetrical and more rounded in occlusal view). This morphology is evident in some of the generally more derived *H. habilis* specimens (OH 13 and OH 16) but is also present in the $M^3$ of KNM-ER 1813, which is otherwise very *Australopithecus*-like. Distal molars are known to be highly variable, while the $M^1$ in particular has a more stable morphology[53,54]. This lower level of variation in the $M^1$ can be considered useful for hominin systematics as it reduces intraspecific variation[55-58], however in this case it seems that differences in EDJ morphology between our samples of *H. habilis* and *Australopithecus* are clearer in the distal molars ($M^2$ and $M^3$). This could suggest that relatively minor morphological differences in the $M^1$s may be exaggerated along the molar row such that they are clearer in the $M^2$ and $M^3$, although this requires further investigation. There was also a significant difference in $P_3$ shape between our samples of *H. habilis* and *Australopithecus*; driven by a slightly increased dentine body height (most extreme in OH 16), symmetrical crown base, and internally placed protoconid dentine horn tip.

There are derived OES dental traits in *H. habilis*, including a buccolingual narrowing of the postcanine dentition[1,41,59] (Supplementary Tables 1 and 2). However, at the EDJ this feature is less clear and is variable by specimen, tooth position, and by which part of the tooth crown contributes to the shape difference. It is possible that differences in the distribution and relative thickness of enamel across the crown contribute to this feature, although this requires further

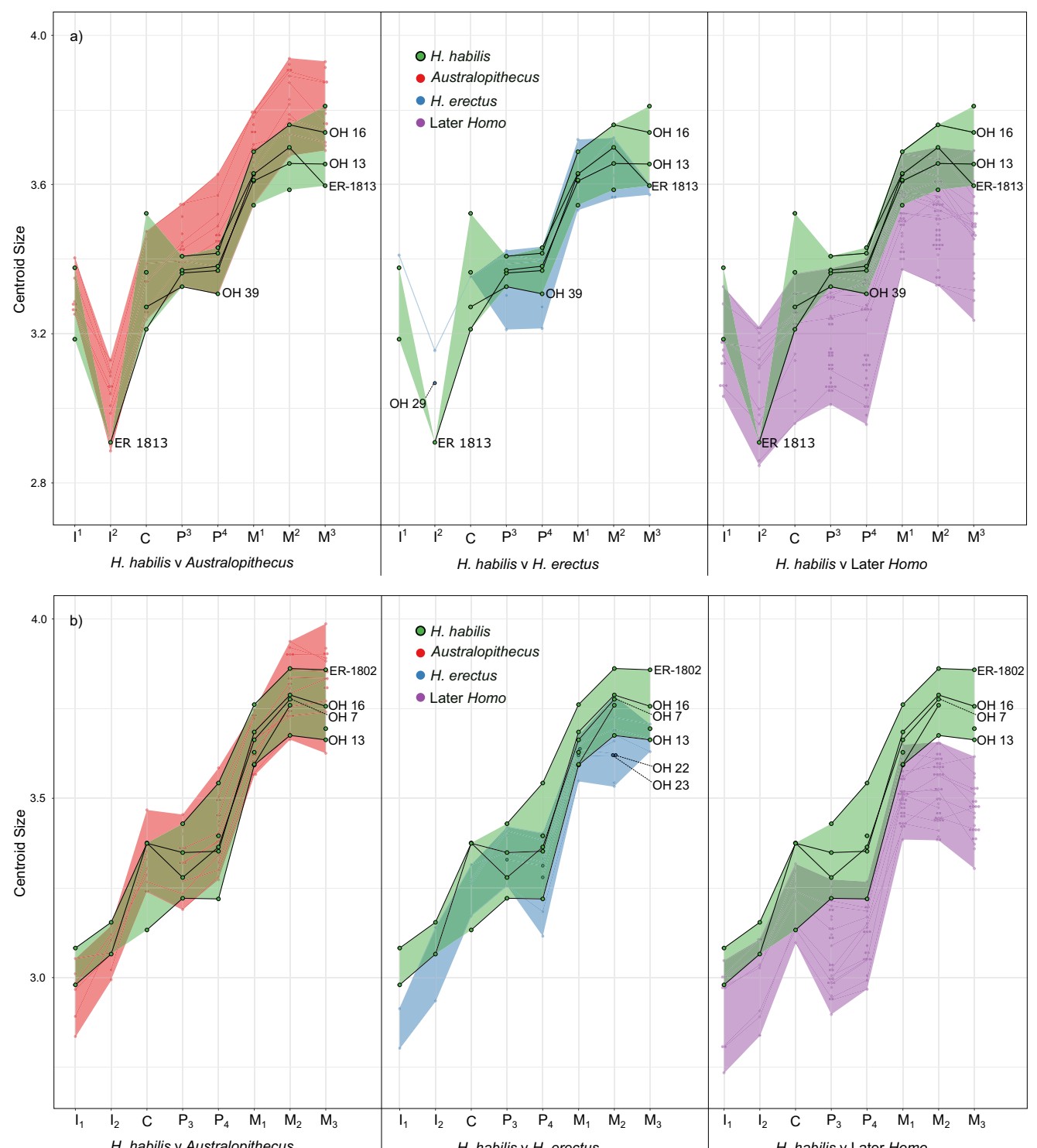

**Fig. 6 | Centroid size across the tooth row.** Plots show the natural logarithm of the centroid size of the CEJ ridge landmark set, calculated separately for **a** the maxillary tooth row, and **b** the mandibular tooth row. In both plots, *Australopithecus* is shown in red, later *Homo* in purple, and *Homo habilis* in green. Shaded areas represent the range of centroid sizes for each group, and lines connect teeth from the same individual. *H. habilis* specimens that preserve complete or nearly complete tooth rows are labelled and connected using black lines. Source data are available in Supplementary Data 21.

investigation. Similarly, we do not find consistent evidence for large anterior teeth in *H. habilis* at the crown base, when compared with *Australopithecus* or later *Homo* (Supplementary Table 6). For a more detailed discussion of both of these traits, see Supplementary Note 1. Quam et al. also found that the M¹ of KNM-ER 1813 and OH 21 show derived relative cusp areas compared with *Australopithecus* and other

*H. habilis* specimens[55]; this is not clear at the EDJ, where both specimens show a primitive overall M¹ shape in PCAs, although the relationship between EDJ morphology and relative cusp areas is likely to be complex, with contributions from multiple aspects of EDJ shape, as well as the thickness and regional distribution of enamel. It is possible that there are additional dental traits not identified here that are

autapomorphic for *H. habilis*, or that effectively distinguish early *Homo* from *Australopithecus*. In particular, this may be the case for aspects of morphology that are not captured in our GM analysis (such as the occlusal basin or lateral faces of the EDJ), or discrete dental traits. However, as shown recently for lower molar accessory cusps, such discrete dental traits frequently show complex expression patterns that can be problematic in studies of hominin systematics[60].

The lack of clearly derived *H. habilis* dental traits at the EDJ in most tooth positions, combined with the overall primitive postcanine EDJ morphology in a number of key *H. habilis* specimens, suggest an overall *Australopithecus*-like endostructural dental morphology in the earliest members of the genus *Homo*. Some of the specimens displaying a more primitive morphology, such as KNM-ER 1813 and OH 24, have been suggested by some authors to belong to *Australopithecus* rather than *Homo*[30–32]. OH 7 is important in this respect, however, as it combines a largely *Australopithecus*-like dental and gnathic pattern with an enlarged endocranial volume that is consistent with other early members of the genus *Homo*[22,61]. A recent study by Zanolli et al.[52] analysed the EDJ morphology of 23 specimens from South African cave sites that have been suggested to belong to the genus *Homo*, finding that only four of the specimens analysed could be unambiguously attributed to *Homo*, while others show a dental pattern clearly more similar to *Australopithecus* or *Paranthropus*. Unlike eastern African specimens such as OH 7, OH 24, and KNM-ER 1813, many of the southern African specimens analysed are isolated teeth, and therefore assessments of their taxonomic status necessarily rely heavily on dental morphology. Since the eastern African Early Pleistocene hominin fossils generally preserve more anatomical structures across the skeleton, future studies will have to integrate all the available morphology: dental, cranial, and postcranial, when assessing the taxonomic affinities of individual hominin specimens.

Alternatively, some have advocated for the entire hypodigms of *H. habilis* and/or *H. rudolfensis* to be transferred out of the genus *Homo*[10,33–36]. They suggest that the inclusion of these species in *Homo* could render the genus paraphyletic, which is supported by some older phylogenetic analyses[33,62]. However, it is important to mention that several more recent phylogenetic studies have supported the monophyly of *Homo* even with the inclusion of *H. habilis/ rudolfensis*[37,63], although in certain instances, maintaining this monophyly might necessitate reconsideration of the generic status of *A. sediba*, *H. naledi* and/or *H. floresiensis*[38,39]. It has also been argued that species included in *Homo* should occupy a coherent adaptive grade[33], and that plesiomorphic features across the *H. habilis* skeleton (e.g., refs. 22,64–66) suggest that the adaptive strategy of *H. habilis* is more similar to *Australopithecus* than *H. sapiens*[33,36]. Others have argued against the requirement for genera to be adaptively unified[16], emphasising that the adaptations seen in *H. sapiens* evolved in a mosaic, stepwise pattern and that the distinction between *Australopithecus* and the earliest members of the genus *Homo* is necessarily slight[15,16]. The results presented here highlight a number of plesiomorphic features across the *H. habilis* dentition that emphasise the species' morphological and adaptive distinction from later members of the genus *Homo*, but whether this justifies removing the species from *Homo* depends on the definition of the genus.

When considering *H. erectus*, there is also some overlap with *Australopithecus* in PCAs for several tooth positions, however, *H. erectus* is better distinguished from *Australopithecus* in CVA analyses, and there are significant differences in EDJ shape between the two taxa in all mandibular tooth positions, as well as $P^4$ and $M^2$. *H. erectus* specimens are distinguished from *Australopithecus* by a number of features, including a reduction of the talon/talonid and a more symmetrical CEJ shape in the premolars (except $P^3$), and more rounded EDJ ridges in the molars. There are also significant differences in size for some posterior molars ($M^3$, $M_2$, $M_3$). Overall, these differences are more pronounced than the differences between *Australopithecus* and

*H. habilis*, and are indicative of a more derived dental morphology in *H. erectus*, even in the Early Pleistocene specimens included here. The small size of the posterior molars is consistent with the finding that *H. erectus* is derived in having relatively reduced $M_3$s (relative to the $M_1$), while early *Homo* specimens are more similar to *Australopithecus*[67] (although the *H. erectus* sample in ref. 67 is mostly younger than the Early Pleistocene specimens included here). One *H. erectus* specimen, KNM-ER 1507, was found to have a more primitive EDJ morphology, with $P_3$-$M_2$ resembling *H. habilis* or *Australopithecus* more than other specimens of *H. erectus*, which is supported by CVA-based classifications (Supplementary Table 7). It is possible that this specimen would be better attributed to *H. habilis*, however, this should be further assessed with reference to the morphology of the mandible, as well as the deciduous molars, particularly as previous attributions of this specimen to *H. erectus* have relied on these aspects of morphology[5,68]. Previous analyses of this specimen have also outlined similarities with another juvenile mandible, KNM-ER 820, which could not be included in EDJ analyses here due to poor tissue distinction in scans. OH 22 shares some similarities with *H. erectus* specimens from Koobi Fora and Sangiran, but is also distinct in features such as the elongation of the mesial fovea (particularly in the $P_3$), and shows more extreme expression of some features seen in early *H. erectus* specimens, such as talonid reduction in the $P_4$ and rounded $M_2$ EDJ marginal ridges. This combination is consistent with the more recent age of the specimen compared to the other *H. erectus* specimens included here, although comparison with younger *H. erectus* specimens would be necessary to fully evaluate the taxonomic affinities of OH 22.

We find a considerable level of variation within the *H. habilis* dental hypodigm that in some cases exceeds that expected of a single species (Figs. 3, 4, and 5; Supplementary Figs. 10 and 11). Bed I Olduvai specimens generally show a more *Australopithecus*-like morphology; OH 7 (-1.84 Ma[69]) and OH 24 (-1.86 Ma[69]) are close to or within the *Australopithecus* range of variation for all preserved tooth positions (Supplementary Fig. 6), while OH 39 (-1.82 Ma[69]) is within the *Australopithecus* range for both premolars but has a more derived $M^2$, and has generally small teeth, some of which are smaller than those of *Australopithecus* ($I^1$, $M^2$). Of the Koobi Fora specimens, KNM-ER 1802 is the oldest at 1.98-2.09 Ma[70], and the postcanine teeth are consistently *Australopithecus*-like in overall morphology, as well as being larger than any other *H. habilis* specimen. Similarly, the teeth of KNM-ER 1813 (1.78–1.95 Ma[71]) are largely primitive in morphology, albeit with a distinct $M^3$ and generally small teeth.

The best-preserved *H. habilis* specimens from Olduvai Bed II are OH 13 and OH 16, both of which preserve entire mandibular and maxillary postcanine tooth rows. OH 13 is from the middle of Bed II and is the youngest specimen assigned to *H. habilis*, with a suggested age of 1.65 Ma[72,73]. OH 16 is older, deriving from the lower section of Bed II. The specimen comes from the above marker Tuff $I^F$, which makes it no older than 1.8 Ma[69,72]. Some early criticisms of the naming of the species *H. habilis* suggested that these Bed II specimens represented a different taxa to the older Bed I material[74,75], while others suggested the Bed II specimens could be distinguished from each other, with either OH 13[76,77] or OH 16[78] more similar to *H. erectus* specimens known at the time. We find that OH 16 is clearly more derived than other *H. habilis* specimens in several aspects of postcanine EDJ morphology, and the specimen displays a suite of derived traits including an increased relative dentine body height in the postcanine dentition, which is typical of later *Homo*. In this regard OH 16 differs significantly from OH 7 and KNM-ER 1802 (lower premolars), OH 39 (upper premolars), and KNM-ER 1813 ($P^3$-$M^2$); these differences in dentine body height exceed the level of variation expected within a species, based on extant hominines (Supplementary Fig. 11). However, it is important to note that there are key differences between OH 16 and later *Homo*, in particular, the specimen retains several features that are clearly primitive (as seen in other

*H. habilis* specimens), particularly in the EDJ marginal ridge shape of the P$_4$, M$_1$, P$^3$, and M$^1$.

The derived features present in the OH 16 dentition could suggest evolution within the *H. habilis* lineage over time. However, despite its younger age, OH 13 is not as derived; The P$_4$, M$_1$, and M$^2$ are similar to OH 16, but for a number of tooth positions such as the P$_3$, the morphology is more similar to Bed I specimens such as OH 7. The specimen therefore shows an intermediate morphology overall. It is possible that the differences between it and OH 16 represent normal variation within the Bed II sample; the differences between the two specimens are within the expected limits for a species based on most extant samples for all tooth positions (Supplementary Figs. 10 and 11). It should be noted that another specimen that is suggested to derive from lower Bed II, OH 21, does significantly differ from the M$^1$ of OH 16 (in Procrustes distance and dentine body height) and is most similar in morphology to KNM-ER 1813. However, there is some uncertainty over the provenance of this specimen, which was found on the surface of a disturbed deposit[41]. Overall, while it is clear that there is substantial variation within specimens attributed to *H. habilis*, arguably too much to be subsumed within a single species, it is not clear to what degree time depth influences this variation, if at all.

The high level of variability in the early *Homo* fossil record has led many authors to suggest that multiple species are represented[17,19–22], although there is little agreement over the arrangement of specimens into groups. Unfortunately, KNM-ER 1470, which is central to this debate, does not preserve any tooth crowns, while the OH 65 maxilla, suggested to strongly resemble KNM-ER 1470[31], could not be included in this sample. Equally, several specimens suggested to belong to this second species of early *Homo* such as KNM-ER 60000, KNM-ER 62000, and KNM-ER 62003[21], could not yet be included in this sample as they have not yet been microCT scanned. KNM-ER 1590, which was included here, has been likened to KNM-ER 1470, although this is mostly based on the large size of the vault[5,79]. We find that the teeth of this specimen are more derived along PC1 than KNM-ER 1813, and the M$^2$ is particularly derived. All teeth are larger in size than those of *H. habilis*, which is consistent with a recent study of canine size that found that including KNM-ER 1590 in a sample of *H. habilis* s.l. led to greater variability than seen in modern humans[80]. The EDJ shape is not as derived as OH 16 in dentine body height, talonid reduction, or a reduction in cusp height, but shows an intermediate morphology in these features. In CVAs, the premolars of KNM-ER 1590 are found to be similar to *Australopithecus*, and classify with the group. The molars show a more mixed signal, with some classifications with *Homo*, but more often not classifying with any group. This would be consistent with the specimen belonging to a second species of non-erectus early *Homo*. However, a further study including some of the specimens mentioned above would be necessary for a detailed dental assessment of this issue. Another important specimen is A.L. 666-1, a 2.3 Ma maxilla from Hadar. The specimen has been attributed to *Homo* aff. *H. habilis*[25], but more recently the arcade shape was found to be a poor match for OH 7 (and therefore *H. habilis*), and instead shows shortened postcanine tooth rows as seen in early *H. erectus* and *H. sapiens*[22]. We find that the premolar morphology of this specimen is quite primitive, with both premolars within the range of *Australopithecus*, but also close to those of OH 39. The premolars also classify as *Australopithecus* in CVA analyses. As in other early *Homo* specimens, the M$^2$ is more derived, showing among other features, a mesiodistally shortened EDJ ridge, and plotting between the *Australopithecus* and later *Homo* groups. The M$^2$ classifies as either *Australopithecus*, *H. habilis*, or none of the groups in CVA analyses, depending on how many PCs are included. The tooth morphology is not found to be similar to the early *H. erectus* specimens included here. This suggests that this specimen combines an overall primitive dental morphology, similar to *Australopithecus* and some other early *Homo* specimens, with a significantly more derived arcade shape. Of the remaining unattributed or *Homo* sp. specimens, there were no clear

attributions for L26-1g (M$_1$), L398-573 (M$^3$), or SK 847 (M$^3$); both M$^3$s occupy an intermediate position in CVA plots, however, the *H. erectus* comparative sample here is only 2 specimens, so this analysis is unlikely to capture the full range of variation in this taxa. However, Omo 75i-1255 (P$_3$) was found to be similar to *H. habilis*, while SK 27 (M$^1$) was found to be most similar to *H. erectus*.

To conclude, we find that the postcanine EDJ morphology of a number of key specimens of *H. habilis*, including the type specimen OH 7, is very similar to that of *Australopithecus*. These results suggest that dental changes associated with later *Homo* were not present in the earliest members of the genus. The EDJ morphology of *H. habilis* is for the most part generalised, which is consistent with the primitive nature of the dental arcade and mandibular corpus in the species[22], as well as important aspects of the postcrania such as the hand morphology[64,65] and relative limb proportions[66]. As such, features linking specimens such as OH 7 to the genus *Homo* are largely limited to the neurocranium. Considering the skeleton as a whole, this indicates that *H. habilis* possessed a number of plesiomorphic features, which is relevant for ongoing discussions over the genus-level designation of *H. habilis*[16,36,81,82]. Finally, our results highlight excessive levels of dental variation within the hypodigm as currently defined; Bed II specimen OH 16 shows a suite of derived dental traits that clearly distinguish it from other *H. habilis* specimens, but do not appear to align the specimen with early African *H. erectus*.

## Methods

### Permissions, outreach, and engagement

Scan data for specimens from Olduvai Gorge was collected for this study after acceptance of a research permit (#2017-182-NA-2016-304) to the Tanzanian Commission for Science and Technology (COSTECH) and under a memorandum of understanding signed between MMS and the National Museum of Tanzania. All of the original CT scan data, a high-end computer workstation (with an installation of the Avizo Software package), and 3D prints derived from these scans were deposited with the museum. A training internship at the University of Kent under the supervision of MMS was provided to AG to become proficient in the processing and analysis of CT scans. Additionally, outreach activities were organised by AG at the museum for local primary school student visits, using custom teaching materials available for download on the Human Fossil Record Archive (https://human-fossil-record.org/index.php?/category/6687). This included custom-designed interactive activity sheets created and funded by MMS focused on human evolution and *Homo habilis* in Tanzania, specifically.

### Study sample

The hominin study sample is summarised in Table 1, and a detailed list of hominin specimens can be found in Supplementary Data 21. The sample consists of 712 hominin teeth (91 incisors, 54 canines, 212 premolars, and 355 molars. Additionally, a sample of extant ape postcanines (*Gorilla gorilla* and *Pan troglodytes*, *n* = 199) was collected in order to compare the levels of variation in *H. habilis* with extant taxa (alongside the *Homo sapiens* specimens included in the main study sample); full details of this sample can be found in Supplementary Data 22.

The hominin sample includes 48 teeth assigned to *H. habilis* from Olduvai Gorge (OH 4, 7, 13, 15, 16, 21, 24, 39, 45; Figs. 1 and 2), and 13 from Koobi Fora (KNM-ER 1502, 1802, 1813; Supplementary Figs. 3 and 4). A full list of specimens preserving whole tooth crowns that are considered here to belong to *H. habilis* is available in Supplementary Table 3; for the small number of specimens that could not be included in the present study, the reasons are outlined in this table.

We also include several specimens that have been suggested to belong to *H. habilis* by some authors but do not have secure species-level attributions (SK 27, SK 847, L26-1g, L398-573, and Omo 75i-1255).

These specimens are separated from the *H. habilis* sample for comparative purposes. Additionally, KNM-ER 1590 is included, which has been suggested to belong to *H. rudolfensis* through its cranial similarities with KNM-ER 1470[5,19]. A.L. 666-1 is a 2.3 Ma maxilla from Hadar attributed to *Homo* aff. *H. habilis*[25], but is here grouped separately as *Homo* sp. KNM-ER 5431 has not been given a genus-level designation but has been suggested to show features resembling *Australopithecus afarensis* or early *Homo*[5,23,83].

The *H. erectus* sample is limited to Early Pleistocene *H. erectus* specimens from Koobi Fora (KNM-ER 806, 807, 992, 1507, 1808, 1812, 3733) and Sangiran (Sangiran 1a, 4, 7-3a-c, 7-10, 7-20, 7-26, 7-27, 7-30, 7-32, 7-53, 7-65, 7-89) as early *H. erectus* provides the most appropriate comparison for the early *Homo* material. KNM-ER 807 has alternately been attributed to *Homo* sp[5]. or *H. erectus*[84], but is included here as *H. erectus*. The EDJ morphology of KNM-ER 1507 appears to differ from other *H. erectus* specimens, and so was excluded from permutation tests and projected into CVA analyses to test its group affinities. OH 22, OH 23, and OH 29 are also assigned to *H. erectus*, but are separated from the main *H. erectus* sample in figures as they are younger than the specimens from Koobi Fora and Sangiran (all are from Bed III or above, <1.14 Ma; refs. 72,85–88). We assess this early *Homo* dental hypodigm against a pooled *Australopithecus* sample, consisting of *A. afarensis* and *Australopithecus africanus*, and a pooled later *Homo* sample, consisting of modern humans, Neanderthals, and Middle-Pleistocene *Homo*. These groups are pooled to allow us to identify the overall EDJ shape changes that have taken place during the evolution of the genus *Homo*, and subsequently identify which of these traits are present in *H. habilis*.

### Terminology

Throughout the manuscript, we refer to the morphology of the EDJ rather than the OES unless otherwise specified. At the EDJ it is clear that the height of the crown can be divided into two components: dentine body height and dentine horn height. Dentine body height refers to the distance between the cervix and the marginal ridges that encircle the occlusal basin, while dentine horn height refers to the distance between the marginal ridges and the tip of the tallest dentine horn (Supplementary Fig. 5). Total crown height is the combination of the two. Unless otherwise specified, these aspects of crown height are described relative to overall crown shape, rather than as an absolute measure.

### Microtomography

Microtomographic scans of the sample were obtained using either a SkyScan 1172 or SkyScan 1173 at 100–130 kV and 90–130 microA, a BIR ACTIS 225/300 scanner at 130 kV and 100–120 microA, or a Diondo d3 at 100–140 kv and 100–140 microA and reconstructed as 16-bit tif stacks with an isometric voxel resolution ranging from 13-45 microns. Seven *H. erectus* teeth (Sangiran 7-3b, 7-3c, 7-20, 7-26, 7-53, 7-65, 7-89) were imaged using neutron microtomography at the ANTARES imaging facility located at the e FRM II reactor of the Technical University of Munich, Germany (for technical details see ref. 89).

### Segmentation

For the majority of specimens, TIFF stacks were filtered using only a mean of least variance filter (kernel size one), or a 3D median filter (kernel size of three) followed by a mean of least variance filter (kernel size of three), implemented using MIA open source software[90]. Filtered image stacks were used to segment enamel from dentine using a seed-growing watershed algorithm employed via a custom plugin employed in Avizo 6.3 (Visualisation Sciences Group), before being checked manually (the custom Avizo 6.3 plugin is not publicly available, but modern versions of Avizo, e.g., Avizo 3D 2022.1, have similarly built-in watershed segmentation modules). In specimens with particularly poor contrast between tissue types, enamel and dentine were segmented using the magic wand tool in Avizo v.8.0 (FEI Visualisation Sciences Group) using interpolation and manual corrections based on information from the filtered and unfiltered image stacks. In regions where enamel and dentine could not be precisely demarcated, interpolation tools were used to extend from the surrounding regions in which better tissue contrast was present. In all segmentations, particular attention is afforded to the marginal ridges, where landmarks are placed, with checks against unfiltered and filtered image stacks if necessary. Once enamel and dentine were segmented, a triangle-based surface model of the EDJ was produced using the unconstrained smoothing parameter in Avizo and then saved in polygon file format (.ply). In some cases, dental wear removes dentine horn tips. In the case of specimens with minimal wear, the missing portion of the dentine horn was reconstructed following a previously published procedure[91]. This methodology is similar to correcting for interstitial wear and involves inferring the shape of the dentine horn tip from the preserved anatomy of the dentine horn. This was restricted to cases in which the wear was less than wear level 3 as defined by Molnar[92], and the observer was confident of the original position of the dentine horn using their experience, anatomical knowledge and preserved EDJ morphology. The reconstruction procedure was completed using the surface modification tools in Geomagic Studio 2014 (https://www.geomagic.com), and through manually adjusting the position of landmarks placed on dentine horn tips in Avizo 6.3. Dentine horn reconstructions in *H. habilis* specimens are available in Supplementary Fig. 6.

### Landmark collection and derivation of homologous landmark sets

3D landmarks were collected in Avizo 6.3 in three distinct sets: 'EDJ main', 'EDJ ridge', and 'CEJ ridge' (Fig. 2). EDJ main consists of fixed landmarks placed on the tips of the main cusps (Mandibular premolars −1) protoconid 2) metaconid; Mandibular molars−1) protoconid, 2) metaconid, 3) entoconid, 4) hypoconid; Maxillary premolars−1) protocone, 2) metacone; Maxillary molars−1) protocone, 2) paracone, 3) metacone, 4) hypocone). EDJ ridge consists of sliding landmarks placed along the marginal ridges encircling the tooth, following the same order as the EDJ main landmarks. Both EDJ sets were collected directly on the EDJ surface models, but in specimens with poor contrast between tissue types, EDJ landmarks were checked against, or in some cases placed directly onto, the unfiltered stack to ensure that they are situated correctly along the marginal ridges. In the CEJ ridge set, landmarks were placed along the cementum-enamel junction (CEJ), beginning at the middle of the buccal face of the tooth (premolars and maxillary molars) or at the mesiobuccal corner of the crown, below the protoconid (mandibular molars), and continuing mesially. These landmarks were placed on an isosurface rendering of the unfiltered TIFF stack, or in cases where the CEJ is not visible on the isosurface due to calculus build-up or the presence of an adjacent tooth, landmarks were placed directly on the unfiltered stack.

A cubic spline function was used to fit a smooth curve through both ridge landmark sets, and the EDJ main landmarks were projected onto the EDJ ridge curve to divide it into sections (two sections in premolars, four in molars). A fixed number of equidistantly spaced landmarks were placed on each section of the curves (in premolars, 20 landmarks were placed in the first EDJ ridge section, and 25 in the second, while in molars, each EDJ ridge section has 20 landmarks. The CEJ ridge set has 40 landmarks all tooth positions). Homologous landmarks were then derived in R using the packages Morpho[93] and princurve[94] using a freely available R-based software routine[95], and following previously published protocols for premolars[42,43] and molars[49,96]. EDJ main landmarks were fixed, while those in EDJ ridge and CEJ ridge sets were treated as semilandmarks and allowed to slide along their curves to reduce the bending energy of the thin-plate spline interpolation function calculated between each specimen and

the Procrustes average for the sample[97]. Sliding was performed twice, with landmarks projected back onto the curve after each step, before landmarks were considered geometrically homologous and converted into shape coordinates using generalised least-squares Procrustes superimposition, which removes scale, location, and orientation information from the coordinates[98–101].

For incisors and canines, the same process was followed, but with landmarks placed around the CEJ only. Landmarks were placed starting on the midpoint of the buccal face and continuing mesially, and as in other tooth positions, 40 equidistant CEJ ridge landmarks were derived, prior to sliding.

## Analysis of EDJ shape and size

Principal component analysis (PCA) was carried out using the Procrustes coordinates of each specimen in shape space. This was completed separately for each postcanine mandibular and maxillary postcanine tooth position using both EDJ and CEJ landmark sets. For the anterior teeth, the same was done using only the CEJ landmark sets. In all cases, PCAs were used as an initial step to explore variation in EDJ shape and to identify EDJ features that distinguish between *Australopithecus* and later *Homo*. These shape changes were visualised using wireframe images of EDJ shape change across principal components. The *H. habilis* specimens were then assessed according to their position in PCA plots, as well as by manually assessing the expression of EDJ features that were identified as important (e.g., those that distinguish *Australopithecus* from later *Homo*).

Canonical variate analysis (CVA) was performed on the *Australopithecus*, *H. habilis*, and *H. erectus* samples, using a small subset of principal components (between 5 and 10) in order to ensure that the number of specimens exceeded the number of variables. In each case, the number of PCs with the highest cross-validated classification accuracy was selected for use in figures, and wireframe images were produced to visualise EDJ shape change across CVs. Cross-validated CVAs (cvCVAs) were also produced in order to check for spurious group separation. The group affinities of specimens without a specific attribution were assessed by projecting them into CVAs before calculating typicality probabilities; this was repeated for each subset of PCs (5-10, resulting in 6 CVAs) in order to ensure that the choice of PC subsets did not influence the results. Group affinities were assessed using a conservative threshold of 0.1; if typicality probabilities for all three groups were below this threshold, the specimen was considered to be unclassified.

The size of specimens was analysed using the natural logarithm of centroid size in two separate analyses, one for mandibular teeth and one for maxillary teeth. Only the CEJ landmark sets were used for these analyses to allow comparisons of size patterns across the entire tooth row.

Two analyses were conducted in order to assess the levels of variation expected within a species. In the first, Procrustes distances were calculated between all possible pairs of individuals within each of the extant species (*P. troglodytes*, *G. gorilla*, *H. sapiens*), as well as between each possible between-group pair for *Pan−Gorilla* and *Pan−H. sapiens*. These were plotted as frequency distributions in order to visualise inter- and intra-group variation and were compared to the differences observed between *H. habilis* specimens (specifically, OH 16 vs other *H. habilis*). This was done separately for each postcanine tooth position (except for the third molars). For the second analysis, a specific aspect of EDJ shape is considered; dentine body height. Dentine body height refers to the distance between the cervix and the occlusal basin of the tooth (equivalent to the total height of the crown minus the height of the dentine horns, also referred to as cervical height[102]). In premolars, a relatively taller dentine body is known to distinguish later *Homo* from *Australopithecus*[42,43], and in modern humans is found to correlate strongly with tooth size. Dentine body height was quantified here by taking the centroid of a subset of landmarks on the mesial and

distal marginal ridges of each tooth, as well as the centroid of a corresponding subset of landmarks on the mesial and distal sides of the cervix, then calculating the distance between the two. The two subsets of landmarks were chosen separately for each tooth position in order to capture the lowest portion of the marginal ridges, between dentine horns, in these specimens. On lower molars, the hypoconulid often occupies a distal position on the crown, so instead two distal subsets were used, one on either side of the hypoconulid. In each case, the same number of landmarks were used in the mesial and distal subsets. This was completed after Procrustes superimposition, and as such describes dentine body height relative to overall crown shape, rather than as an absolute measure. Within- and between-group differences in dentine body height were then calculated and visualised in the same manner as described above for Procrustes distances.

We also tested for shape and size differences between groups using permutation tests, carried out in R using 10,000 permutations, separately for each postcanine tooth position. For shape, this was done using Procrustes coordinates from the full EDJ analysis, and for size, this was done using centroid size from the CEJ analysis to allow the inclusion of anterior teeth. The Benjamini-Hochberg procedure was used to control the false-discovery rate[103].

## Reporting summary

Further information on research design is available in the Nature Portfolio Reporting Summary linked to this article.

## Data availability

The geometric morphometric landmark data generated in this study have been deposited in the Publications section of The Human Fossil Record (https://human-fossil-record.org/index.php?/category/14230). The raw scan data used in this study are curated by the museums and institutes that curate the original fossil material. These data were used under a MOU for the current study, and so are not publicly available, but can be accessed by research application to the relevant curatorial institution (see Supplementary Data 21). The source data for Figs. 3, 4, 5, and Supplementary Figs. 7, 10, 11, 12, 13, 14, 15, and 16 are provided as a Source Data file. The source data for Fig. 6 and Supplementary Fig. 17 are available in Supplementary Data 21. Supplementary Tables 1 and 2 are based on previously published data, which is available in references listed in Supplementary Note 1. Source data are provided in this paper.

## Code availability

R code used for geometric morphometric analyses is available from: https://zenodo.org/doi/10.5281/zenodo.10255288[94].

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

## Acknowledgements

For access to specimens and microCT scans, we would like to thank Emma Mbua and Samuel Muteti (National Museums of Kenya), Audax Mabulla (National Museum of Tanzania), the Tanzania Commission for Science and Technology, Metasebia Endalemaw, Yared Assefa (Ethiopian Authority for Research and Conservation of Cultural Heritage), Bernhard Zipfel, Sifelani Jira (Evolutionary Studies Institute, University of the Witwatersrand), Miriam Tawane (Ditsong Museum), Davorka Radovčić (Croatian Natural History Museum), Michel Toussaint (ASBL Archéologie Andennaise, Royal Belgian Institute of Natural Sciences), Jean-Jacques Cleyet-Merle (Musée National de Préhistoire des Eyzies-de-Tayac), Véronique Merlin-Langlade (Musée d'Art et d'Archéologie du Périgord), Christine Feja (Institut für Anatomie der Universität Leipzig), Antonio Rosas (Museo Nacional de Ciencias Naturales), Flora Gröning (Institute of Medical Sciences, University of Aberdeen), Dejana Brajković (Croatian Academy of Sciences and Arts), Jean-François Tournepiche (Musée d'Angoulême), Patrick Périn (Musée d'Archéologie nationale de Saint-Germain-en-Laye), Paul Tafforeau (European Synchrotron Radiation Facility), David Hunt (National Museum of Natural History), Ullrich Glasmacher (Institut für Geowissenschaften, Universität Heidelberg), Mirjana Roksandic (University of Winnipeg), Dušan Mihailović, Bojana Mihailović (Department of Archaeology, Belgrade University and the National Museum), Eleni Panagopoulou (Ephorate of Palaeoanthropology & Speleology of Southern Greece), Katarina Harvati (University of Tübingen), Martin Fischer (Phyletisches Museum Jena), Irina Ruf, Friedemann Schrenk and Christine Hemm (Senckenberg Forschungsinstitut und Naturmuseum, Frankfurt), Frieder Mayer and Christiane Funk (Museum für Naturkunde—Leibniz Institute for Evolution and Biodiversity Science), Emmanuel Gilissen (Royal Museum for Central Africa, Tervuren), Christophe Boesch and Roman Wittig (Taï Chimpanzee Project), and the Senckenberg, Forschungsstation für Quartärpaläontologie. We express our gratitude to the Werner Reimers Foundation in Bad Homburg (Germany), which provides the Gustav Heinrich Ralph von Koenigswald collection as a permanent loan for scientific research to the Senckenberg Research Institute and Natural History Museum Frankfurt. For µCT-scanning and technical assistance, we thank Kudakwashe Jakata, David Plotzki, and Heiko Temming. We thank the ANTARES imaging facility located at the FRM II reactor of the Technical University of Munich, Germany and Burkhard Schillinger for the possibility of neutron scanning of the specimens from Sangiran. For segmentation assistance, we thank Lia Schurtenberger, Rhianna Drummond-Clarke and Mykolas Imbrasas. This project has received funding from the Max Planck Society (T.W.D., P.G., J.-J.H.), the European Research Council (grant agreement No. 819960; M.M.S.) and the Calleva Foundation (U.K.; F.S.).

## Author contributions

T.W.D., P.G., F.S., Z.A., A.G., J.-J.H., W.K., O.K., W.P., C.Z., and M.M.S. designed research and wrote the paper; T.W.D., P.G., F.S., W.P., C.Z., M.M.S. performed research and analysed data; F.S., Z.A., A.G., J.-J.H., W.K., M.M.S contributed to microCT scanning of fossils.

## Funding

## Competing interests

The authors declare no competing interests.
