## [Peer Review File · Nature Communications]

Dental morphology in *Homo habilis* and its implications for the evolution of early *Homo*Reviewers' Comments:

Reviewer #1:

Remarks to the Author:

Thank you for asking me to review this very interesting paper written by Davies and colleagues.

In this manuscript, the authors assess the EDJ morphology of some 12 key specimens from Olduvai (n= 9) and from Koobi Fora (n=3). Using geometric morphometry bases on 3D landmarks , they characterize the main difference in EDJ shape between Australopithecus (A. afarensis and A. africanus) and modern humans, Neandertals, and then identify which of these dental features are present in the Homo habilis specimens studied in this paper (which come from two sites from East Africa) in order to discuss the homogeneity of the Homo habilis hypodigm.

The authors also aim to address the EDJ manifestation of some traits previously considered characteristic of H. habilis and to discuss the homogeneity of the H. habilis hypodigm.

One of the main conclusions of this paper is that the postcanine EDJ morphology of the Homo habilis specimens studied is very similar to that of Australopithecus. The authors also highlight excessive levels of dental variation within their sample, particularly with the specimen OH 16 from Olduvai.

This original study on the eastern african specimens allocated to Homo habilis is of great importance for the discussion of the beginning of our evolutionary history, since Homo habilis is considered, along with Homo rudolfensis, as the oldest species of our genus.

With respect to the validity and robustness of the nterpretation and conclusion of the data, some revisions are necessary (see below some general remarks, followed by more specific remarks).

General remarks

The title should be changed because only some specimens from East Africa (Kenya and Tanzania) attributed to Homo habilis are included in this analysis, and no specimens from South Africa. This should also be the case whenever the results on Homo habilis are mentioned, especially in the conclusion and introduction.

In my opinion, it might be interesting to include specimens from South Africa (which means that the title will be relevant in this case), as the data have already been collected and analyzed by some authors of this publication (including the first author e.g. Davies et al., 2021; Zanolli et al., 2022), even though it is mentioned in the paragraph "study sample" that "Proposed early Homo specimens from southern Africa are not included here as the majority do not have a secure species level attribution, and the inclusion of many of these specimens within the genus Homo has been called into question". In addition, it might be interesting to include some specimens previously studied by the first author (Davies et al., 2021) from the Omo valley and Kooki Fora.

The authors should put in the supplementary information, the hypodigm that they consider to be Homo habilis (as the hypodigm can differ between different authors), and in bold the specimens included in their current analysis. In fact, the authors discuss the samples they studied but not the hypodigm of Homo habilis (which includes more specimens). This nuance should also be made in the manuscript, especially when the authors mention that they are discussing the homogeneity of the hypodigm.

A more thorough discussion between the results for Homo habilis and Homo erectus in relation to the Australopithecus specimens should be added, in particular to see if they differ much, especially since it is mentioned in the title "its implication for early Homo evolution". Regarding this last point, this implication should be further developed in the discussion.

In the figures 3 and 5. The Homo erectus specimens are also close to Australopithecus, such as Homo

habilis (except OH 16 in figure 3). This should be discussed in the manuscript.

Regarding the methods, as one of the objectives was to examine distinctions between groups, why did the authors not use between group PCA (bg PCA), because this method was developed to summarize difference between groups in high-dimensional space as in geometric morphometrics. Furthermore, this method has been used previously by some of the co-authors in some papers (Zanolli et al., 2022 for example), where the objective was quite similar ("Dental data challenge the ubiquitous presence of Homo in the cradle of Humankind").

The authors mention the discrete traits. I wonder why the authors did not include and score them in this paper, given that the first author has already published a paper regarding the lower molar and shows that there are taxon-specific patterns, although we must be cautious in comparisons of accessory cusps for taxonomic interpretations.

The inclusion of some discrete traits in OES and EDJ will be very interesting and could be use later by some researchers for phylogenetic analyses.

Specific comments

Abstract

L5 and L9. Add east African before H. habilis... if the south african specimens are not included in the next version of the manuscript.

L 35. More references should be added from West Turkana (WT 42718) and from the Omo valley.

L41. The reference of Leakey et al., 2001 should also be included as they state that "The transfer of this species (i.e. rudolfensis) to Australopithecus has been recommended, but Kenyanthropus may be a more appropriate genus".

L41. It should also be mentioned that some studies (based on the same type of analysis (phylogenetic analyses), do not reach the same conclusions (Strait et Grine, JHE 2004; Dembo et al., JHE 2016; Caparros and Prat, IScience 2021). This comment and references should be added in the manuscript.

L42. What is the proportion of dental remains? by mentioning "the large portion".

L76. Since specimens from southern Africa attributed to early Homo are not included in this study (see more general remark), "specimens of H. habilis" should be replaced by "some H. habilis specimens from eastern Africa".

L78. If one of the goals of this paper is to assess the homogeneity of H. habilis hypodigm, the authors should mention in the SI which hypodigm they considered (i.e. specify which specimens are included in H. habilis). Since this study is only conducted in one part of the hypodigm, due to the study of dental remains and only some from Olduvai and Kooki Fora, this sentence is bit exaggerated and goes much further than what can be argued in this paper.

L83. Method. Why the authors did not used bg PCA (see general remark)?

L86. It should be important in all the analyses to distinguish A. africanus and A. afarensis (two different ellipses), as these species originate probably from two different regions with possibly different evolutionary histories.

For the purposes of figures, analyses and interpretations, it would be good to plot (and name) all Homo erectus specimens (which are in fact african Homo erectus, as there is no asian Homo erectus in the sample), instead of putting H. erectus with a line (Figure S6, Figure 3..).

In Figure 3. There seem to be two specimens close to OH 16, which are very different from the other

specimens). What are the labels of these specimens?

L 93. Figure S5 illustrates the term used to describe crown height at the EDJ, but not why the increase of the height of the dentine body is the notable shape between *Australopithecus* and later *Homo*. Figures and analyses that support this conclusion should be included here.

Table 2. It would be interesting to include in this table clearly the shape characteristics for the african *Homo erectus* specimens, and for *Australopithecus afarensis* and *Australopithecus africanus*.

L 102. It would be interesting to include in the supplementary informations, the condition (cf. Table 2) of all the *Homo habilis* studied here (including the Kooki fora specimens).

L104. How many specimens of *Homo habilis* retain the primitive conditions?

L116. Could you give the number of specimens and their accession number, instead of "several". *Homo erectus* should be changed by african *Homo erectus*, as there is no asian *Homo erectus* in the sample.

L 119. Since some african *Homo erectus* specimens are mentioned in the text to be distinguished from *Australopithecus*, their label (name) must appear clearly in all PCA Figures.

L152. I am not sure if this sentence refers to Supplementary Fig.S3?

L 175. In Figure 6, it would be good to make the distinction between *A. africanus* and *A. afarensis* and to label in this figure AL 199-1 and AL 200-1a, as they are mentioned in the text.

L180. I am not sure if this sentence refers to Supplementary Table S3?

L199. I am not sure if this sentence refers to Supplementary Table S3?

L 202. Figure S3. It might be also interesting to make the same figures with *H. erectus* vs. *Australopithecus* and *H. erectus* vs. Later *Homo*. Since the centroid size across the tooth row of *Homo erectus* is close to that of *H. habilis*, one would expect *H. erectus* to be included in the range of variation of *Australopithecus*. The results of these two graphs concerning *H. erectus* should be therefore discussed.

L203. As KNM-ER 992, 1507 and KNM-WT 15000 are mentioned in the text, their labels should be seen in Supplementary Figure S13.

DISCUSSION

It might also be interesting to discuss whether or not african *Homo erectus* specimens also retain some *Australopithecus* conditions.

L218. Since it is only a part of the *Homo habilis* hypodigm that has been studied here (e.g. no south African specimen, no specimens from the Omo valley and only dental material), it should be more appropriate to be more specific/less general in purpose. Write differences between the *Homo habilis* specimens studied here (mainly from Olduvai and Koobi Fora) instead of differences between *Homo habilis*..., unless, in the revised version, other specimens, especially those from South Africa, are included.

L221. Referring to Table 1, add that the sample size of Upper M3 in this study for *Homo habilis* is n=3. In the discussion, the results presented here should be placed in a broader perspective, including results based on phylogenetic analyses that include some *Homo habilis* and *Australopithecus* specimens using craniodental traits (Strait et al., JHE 2004; Dembo et al., JHE 2016; Caparros and Prat, IScience 2021) or dental traits (Irish et al., Science, 2013).

L238. To make this sentence more precise, could the authors list the dental traits, and if necessary

discuss them with the literature.

L256. The authors mention the discrete traits. I wonder why the authors did not include and score them in this paper, given that the first author has already published a paper regarding the lower molar and shows that there are taxon-specific patterns, although we must be cautious in the comparisons of accessory cusps for taxonomic interpretations.

The inclusion of some discrete traits in OES and EDJ will be very interesting and could be use later by some researchers for phylogenetic analyses.

Furthermore, this list of discrete traits and results could be discussed with the results from Irish et al., where *habilis/rudolfensis* are related to *H. erectus* and not to *A. africanus*.

L 264. Reference 48 (Leakey et al., 1974) should probably be deleted, as the number of specimens was very different at that time and as mentioned by Leakey et al., 1974 themselves, "A second genus would incorporate many of the gracile specimens from Sterkfontein presently referred to *A. africanus*, perhaps certain specimens from East Rudolf including KNM-ER 1813, and possibly some from Olduvai such as OH 24 [...] These remarks are necessarily speculative. A more detailed review of hominid systematics is being prepared in collaboration with B. A. Wood.. " and B. Wood did not include these specimens in *Australopithecus* in the monograph concerning Koobi Fora in 1991.

L280. Add east african before *H. habilis*, as the sample studied here is not the hypodigm of *H. habilis*. In this part of the discussion, a paragraph about east African *H. erectus* should be added, and especially when OH 16 and OH 13 are mentioned.

Conclusion

L 355. "suggesting that dental changes that are associated with later species of the genus *Homo*". Is it also true for *Homo erectus*, or does this sentence refer only to the later *Homo* sample?

L359-L 363. This part should probably be discussed in the discussion paragraph and placed in a more general perspective including results based on phylogenetic analyses that include *Homo habilis* and *Australopithecus* specimens (Strait et al., JHE 2004; Dembo et al., JHE 2016; Caparros and Prat, IScience 2021, Irish et al., Science 2013) (see comments on the discussion).

L364. The authors mention that their results highlight excessive levels of dental variation of the *H. habilis* hypodigm, but this study concerns only one part of the hypodigm. It would be nice if the authors would mention in the supplementary information, what they consider to be the *Homo habilis* hypodigm (see also my previous comment in general remark).

Supplementary informations.

Supplementary note 1.

I am not sure if autapomorphy is the most appropriate term. How do you test it? How are you sure that the traits mentioned are autapomorphies (taxon speciifc) and not synapomorphies or symplesiomorphies?

Figure S9, S10 and Figure 5. What are the distributions for *H. erectus*, Neandertals, Modern humans, *A. africanus* and *A. afarensis*.

Supplementary tables.

Table S1 and Table S2. It is worth mentioning where the data come from. From this study? If not (which is probably the case, as there are sometimes more specimens in Table S1 than in Table 1), could the authors add the reference.

Reviewer #2:

Remarks to the Author:

This study conducts a large-scale analysis of enamel-dentine junction morphology in *Homo habilis*, with a large comparative sample of fossil hominins and extant apes. The paper is well-written with refined figures and clear language throughout. Their analyses focus on variation in EDJ morphology in *Homo habilis* using micro-CT scans and Procrustes analyses. All methods employed are appropriate. Using EDJ variation across several different teeth, the authors find that OH 16 is more similar to later *Homo*, while most other specimens attributed to *Homo habilis*, including OH 7, the holotype, look much more like *Australopithecus*. Recent publications investigating dental morphology in hominins have also found evidence that the dental morphology of early *Homo* looks very *Australopithecus*-like (e.g., Brasil et al. 2020; Jiménez-Arenas 2020).

Overall, this is a very strong study with interesting findings that are relevant to human evolutionary studies and paleoanthropology. I think the paper is well written and well-presented, and the results support the interpretation. However, I have three main concerns with the manuscript which I feel require addressing.

1) First and most importantly, I have serious concerns about the data availability for this project. As currently presented, readers are unable to assess data reliability. Additionally, the study does not offer replicability, a key requirement of all science. As written, the data availability statement does not meet the requirements of the journal. If scan data cannot be made directly available due to concerns by owning nations or repositories, these conditions should be discussed directly in the data availability statement. Under what conditions can scans be accessed? How did the authors access these scans? Additionally, the exact requirements required for access to landmarked datasets should be included in the data availability statement. Far too many researchers have reported being denied access to data despite data availability statements claiming access will be granted "upon reasonable request". Please indicate exactly what a reasonable request is, so that future researchers wishing to replicate and build upon this research have transparency about the location and availability of these data. The goal of science is to build up, not to hold up. If these data are ready for publication, then they are ready to be made available for scientific purposes. Please address all journal requirements and revise your data availability and/or statement.

JOURNAL REQUIREMENTS

An inherent principle of publication is that others should be able to replicate and build upon the authors' published claims. A condition of publication in a Nature Portfolio journal is that authors are required to make materials, data, code, and associated protocols promptly available to readers without undue qualifications.

Data availability subject to controlled access: the data availability statement should include the following information: reasons for controlled access (eg., privacy, ethical/legal issues), conditions of access must be described precisely including contact details for access requests, timeframe for response to requests, restrictions imposed on data use via data use agreements. A copy or link to the data use agreement should be provided if requested by editors. Restrictions on controlled access datasets including restrictions on downstream data reuse or authorship requirements must be clearly described in manuscript and to editors at the time of submission. Editors may decline further consideration of the manuscript after evaluation if restrictions are found to be unduly prohibitive.

Third party data: when data obtained from third parties cannot be made available, the restrictions should be clearly stated in the data availability statement. Authors must make data available for purposes of peer review, if requested by reviewers, within the terms of a data use agreement and if compliant with ethical and legal requirements.

2) A survey of the literature cited indicates that the authors have not cited anything that has been released in the last 5 years, unless it was written by them. This is an oversight of the landscape of contemporary paleoanthropology. As such, the authors have missed key citations from scientists active

in the field, many of which play directly into the authors' arguments and analyses. Please see bullet list of a few examples below. I recommend citing these relevant, modern publications in your revisions:

Bruner and Beaudet (2023) – discussion on endocranial variation in *H. habilis*
<https://doi.org/10.1016/j.jhevol.2022.103281>

Brasil et al. (2020) – dental data place *H. habilis* and other early *Homo* as more similar to *Australopithecus*
<https://doi.org/10.1007/s00114-020-01696-9>

Grine et al. (2019) – KNM-ER 64060, affiliation with *H. habilis* -
<https://doi.org/10.1016/j.jhevol.2019.03.017>

Monson et al. (2020) – evolution of EDJ in humans. Specifically discusses dentine body height variation
<https://doi.org/10.1073/pnas.2008037117>

Jiménez-Arenas (2020) – dental research on variability in *H. habilis* with suggestion that it exceeds one species.
<https://doi.org/10.7203/sjp.27.1.18090>

Pan et al. (2020) – EDJ variation and taxonomy in fossil apes
DOI: 10.16359/j.cnki.cn11-1963/q.2020.0023

Another good citation to include is this review of early *Homo* taxonomic discussions by Antón (2012):

Antón (2012) – taxonomic instability of *H. habilis*
<https://doi.org/10.1086/667695>

3) In the methods, the authors mention that they reconstructed several specimens prior to analysis (lines 433-435). Since we are not able to see all specimens, please indicate in table or text all specimens for which reconstructions were performed.

Reviewer #3:

Remarks to the Author:

This paper presents an analysis of the enamel dentin junction of a large sample of hominin teeth to assess the morphology of the EDJ in *Homo habilis*. I found this manuscript to be very well written and methodologically sound. There were certain elements of this paper that I particularly appreciated, such as the inclusion of Table 2 which provides a nice summary of morphological differences between the various hominin groups rather than leaving the reader to interpret undefined shape space on their own.

Despite these strengths, however, I do think one key element is conspicuously missing. One of the stated goals of this paper was to evaluate the homogeneity and affinities of the *H. habilis* hypodigm. Given that *H. habilis* has been reconstructed as sharing its closest phylogenetic affinities to *Homo rudolfensis*, as well as the ongoing controversy over the *H. habilis* vs. *H. rudolfensis* hypodigms, this taxon represents an obvious comparative sample for the questions at hand—arguably much more so than the very large Neanderthal sample which has been included in the study. Lines 331-333 note that several *H. rudolfensis* specimens (OH 65, KNM-ER 60000, KNM-ER 62000, and KNM-ER 62003) could not be included in the study, but give no further explanation. I can probably read between the lines vis

a vis including OH 65, but am confused as to why KNM-ER 60000, KNM-ER 62000, and KNM-ER 62003 could not be included given that one of the key authors who originally described these specimens is also an author on the present study.

On a related note, I see that there is a very large *A. africanus* comparative sample and wonder whether the authors might consider including *A. sediba* as well. Given its purported links to *Homo* (Berger et al., 2010), and specifically to *H. habilis* (e.g., Dembo et al., 2015), this would be an interesting sample to address in the present study.

Another issue that needs to be addressed in the introduction of the paper is the definition of the *H. habilis* hypodigm. Lines 78-79 state that one of the goals of the paper is “to assess the homogeneity of the *H. habilis* hypodigm as currently defined.” Given that this is a controversial taxon, it would be useful to include references here (e.g., as currently defined by whom?) For example, several studies have included Stw 53 in their *H. habilis* hypodigm (Strait and Grine, 2004; and by extension: Dembo et al., 2015; 2016; Mongle et al., 2019; 2023). Was the hypodigm in this manuscript restricted to only include specimens from eastern Africa, do the authors disagree with the hypodigm used by other studies, or was Stw 53 excluded due to its high degree of dental wear? Likewise, Wood (1992) has included other specimens like KNM-ER 1805 in *H. habilis*, but that has not been included here. Was this specimen excluded because the KNM-ER 1805 dentition was considered too damaged to assess or do the authors disagree with Wood (1992)? I would presume that the the authors are following the *H. habilis* hypodigm suggested by Spoor et al. (2015), but this should be explicitly stated.

While I felt that the majority of the study was methodologically sound, I do have some concerns related to the reconstruction of cusp tips for landmark placement in fossil specimens with moderate wear (e.g., OH 7, OH 13, OH 16, AL 666, ER 1813 M1 protoconids). The authors state that these missing portions of the mesh were reconstructed in Geomagic following a protocol established in the senior author’s dissertation. In my own experience, these reconstructions can vary wildly based on the preserved portions of the cusp and the specific Geomagic setting that was used for interpolation. Given that relative protoconid height was one of the characteristic features described for the *H. habilis* dentition (Table 2), it would be useful for readers to be able to evaluate figures of these reconstructed meshes in the supplemental material.

On lines 353-356, the authors conclude: “We find that the postcanine EDJ morphology of a number of key specimens of *H. habilis*, including the type specimen OH 7, is very similar to that of *Australopithecus*, suggesting that dental changes that are associated with later species of the *Homo* genus, were not present in the earliest members of our genus.”

While this conclusion appears to be supported by the one PCA provided in the main text (Fig. 5a), the picture becomes a bit more nuanced when considering results across the full dataset in the supplementary information. For example, the lower M1-M3 PCAs (Fig S6) also show *H. erectus* completely overlapping with *Australopithecines* in morphospace. Had *H. rudolfensis* been included, I would imagine it would find itself overlapping those polygons as well. All of this to say, I think your results speak to a clear difference between Humans/Neanderthals and earlier members of the genus *Homo*, but it is an overstatement of the results to claim that these results indicate “future research should address the implications of this for the genus level designation of the *H. habilis* hypodigm” (lines 362-363)

Table S4: need to indicate what * next to species designation means in table key (e.g., OH 22, OH 23)

We would like to thank all three reviewers for their helpful comments and suggestions. We have significantly revised the manuscript in response to these comments. In particular, we have expanded the sample of early Homo and Homo erectus specimens. Consequently, we have updated all GM analyses of postcanine EDJ shape and centroid size analyses for mandibular and maxillary teeth. As a result of rerunning these analyses, we have updated Figures 3, 4, 5 and 6 and Table 1, as well as Supplementary Figure S7, 8, 9, 15, 17 and Table S6. Figure 6, Table 2, Supplementary Figure S17 and Table S6 have also been updated in response to reviewer comments, and additional tables and figures have been added to the supplementary information (Figures S6, 12, 13, 14, and tables S3, 4, 5 and 7). Details on these changes are outlined below in response to reviewer comments.

We also recomputed the permutation test after expanding the sample and identifying an error related to the method used for calculating pairwise tests in R. The updated results are presented in Supplementary Table S6. While the new results align well with our previous results, there are now additional statistically significant mean differences in: Australopithecus v later Homo (C_1 size), H. habilis and Australopithecus (M^2 , M^3 , P_3 shape), and H. habilis v later Homo (P^3 , M^2 , P_4 , M_3 shape and M^1 , M^2 size). It is important to note, though, that these new results do not alter the overall conclusions of our study. However, to reflect these updates, we have made necessary amendments to the main text.

Please find below detailed responses to each reviewer comment, as well as further details on the revisions we have made to the manuscript.

Reviewer #1

Thank you for asking me to review this very interesting paper written by Davies and colleagues.

In this manuscript, the authors assess the EDJ morphology of some 12 key specimens from Olduvai (n= 9) and from Koobi Fora (n=3). Using geometric morphometry bases on 3D landmarks, they characterize the main difference in EDJ shape between Australopithecus (A. afarensis and A. africanus) and modern humans, Neandertals, and then identify which of these dental features are present in the Homo habilis specimens studied in this paper (which come from two sites from East Africa) in order to discuss the homogeneity of the Homo habilis hypodigm.

The authors also aim to address the EDJ manifestation of some traits previously considered characteristic of H. habilis and to discuss the homogeneity of the H. habilis hypodigm.

One of the main conclusions of this paper is that the postcanine EDJ morphology of the Homo habilis specimens studied is very similar to that of Australopithecus. The authors also highlight excessive levels of dental variation within their sample, particularly with the specimen OH 16 from Olduvai.

This original study on the eastern african specimens allocated to Homo habilis is of great importance for the discussion of the beginning of our evolutionary history, since Homo habilis is considered, along with Homo rudolfensis, as the oldest species of our genus.

With respect to the validity and robustness of the interpretation and conclusion of the data, some revisions are necessary (see below some general remarks, followed by more specific remarks).

General remarks

The title should be changed because only some specimens from East Africa (Kenya and Tanzania) attributed to *Homo habilis* are included in this analysis, and no specimens from South Africa. This should also be the case whenever the results on *Homo habilis* are mentioned, especially in the conclusion and introduction.

In my opinion, it might be interesting to include specimens from South Africa (which means that the title will be relevant in this case), as the data have already been collected and analyzed by some authors of this publication (including the first author e.g. Davies et al., 2021; Zanolli et al., 2022), even though it is mentioned in the paragraph “study sample” that “Proposed early *Homo* specimens from southern Africa are not included here as the majority do not have a secure species level attribution, and the inclusion of many of these specimens within the genus *Homo* has been called into question”. In addition, it might be interesting to include some specimens previously studied by the first author (Davies et al., 2021) from the Omo valley and Kooki Fora.

*Unfortunately, the majority of early *Homo* specimens from South Africa and the Omo valley are not securely attributed to species level, which initially led to their exclusion from our sample. Moreover, several micro-CT scans did not show a clear tissue contrast between enamel and dentine. Nonetheless, we acknowledge the significance of broadening the sample's geographical range. As such, we have included several specimens that have been previously classified as *Homo habilis*: for specimens from South Africa, we limited this to specimens found by Zanolli et al. 2022 to likely belong to *Homo* on the basis of EDJ morphology (so as to avoid repeating analyses). As such we have added SK 27 and SK 847 to our sample. For Omo specimens, we are largely restricted by our ability to distinguish enamel from dentine in micro-CT scans of the fossil material, which is very limited. We have been able to include three specimens; L26-1g, L398-573 and Omo 75i-1255. Three Omo specimens included in the *H. habilis* sample of Davies et al. (2021) were not able to be included here due to excessive tooth wear (Omo 75s-1969-15) and poor preservation of the cervical region (Omo K7 1969-19 and Omo 75s-1969-16). Further Koobi Fora specimens from Davies et al. (2021) do not have secure species level attributions and therefore were not included here.*

*The newly added specimens are separated from the main *H. habilis* sample in figures and statistical analyses, and their attributions are assessed through canonical variate analysis (CVA), which was added to the manuscript to address this and other comments (see below). The specimens were projected into CVAs, and typicality probabilities were calculated to assess their group affinities (Supplementary Table S7).*

The authors should put in the supplementary information, the hypodigm that they consider to be *Homo habilis* (as the hypodigm can differ between different authors), and in bold the specimens included in their current analysis. In fact, the authors discuss the samples they studied but not the hypodigm of *Homo habilis* (which includes more specimens). This nuance should also be made in the manuscript, especially when the authors mention that they are discussing the homogeneity of the hypodigm.

*We have added a table to the supplementary information with a list of specimens preserving tooth crowns that we consider to be attributable to *H. habilis* (Table S3), and the attribution references. In the case of specimens that were not able to be included in our sample, we have listed why this was the case.*

A more thorough discussion between the results for *Homo habilis* and *Homo erectus* in relation to the *Australopithecus* specimens should be added, in particular to see if they differ much, especially

since it is mentioned in the title "its implication for early Homo evolution". Regarding this last point, this implication should be further developed in the discussion.

In the figures 3 and 5. The Homo erectus specimens are also close to Australopithecus, such as Homo habilis (except OH 16 in figure 3). This should be discussed in the manuscript.

In order to give more thorough attention in the manuscript to the differences between H. erectus and H. habilis, and how they both are similar to and differ from Australopithecus, we have taken several steps:

- 1) We have expanded the H. erectus sample to include 21 specimens from Sangiran – this boosts the sample size and increases the geographical range of the sample, but still allows the sample to be restricted to early Pleistocene specimens, which are most relevant for discussions of early Homo.***
- 2) We have added CVAs (made possible by the increased sample size for H. erectus) which allow for a more thorough investigation of the differences in EDJ shape between Australopithecus, H. habilis and H. erectus. The results are shown in Supplementary Figure S12. We also added cross-validated CVAs (cvCVAs) to check for spurious group separation, which can be an issue with such analyses, and the results of this are shown in Supplementary Figures S13 and S14.***
- 3) The increased sample size of H. erectus also allows their inclusion in permutation tests, which are updated in Supplementary Table S6.***
- 4) In response to a comment below, we have replicated Table 2 (which outlines several key EDJ differences between Australopithecus and later Homo, then lists the expression of these features in H. habilis specimens) for several African H. erectus specimens.***

These changes have allowed us to investigate the EDJ morphology of H. erectus in a similar level of detail to H. habilis, and we have therefore expanded the results section (line 202-229) and discussion (line 401-431) of H. erectus EDJ shape.

Regarding the methods, as one of the objectives was to examine distinctions between groups, why did the authors not use between group PCA (bg PCA), because this method was developed to summarize difference between groups in high-dimensional space as in geometric morphometrics. Furthermore, this method has been used previously by some of the co-authors in some papers (Zanolli et al., 2022 for example), where the objective was quite similar ("Dental data challenge the ubiquitous presence of Homo in the cradle of Humankind").

In light of recent methodological critiques of between-group PCA (see Cardini et al. 2019; Bookstein 2019, Evolutionary Biology), we decided not to use this method. In short, these authors show that bgPCA can lead to spurious clustering (see also Cardini and Polly 2020). Instead, we have added canonical variate analysis (CVA) to investigate group differences, also used by Zanolli et al. (2022). To avoid spurious group separations similar to the bgPCA pathology mentioned above, we performed the CVA on a lower dimensional subspace using principal component analysis to reduce the dimensionality of the original data. In each case, the number of PCs with the highest cross-validated classification accuracy was selected for use in figures, and wireframe images were produced to visualise EDJ shape change across CVs. Cross validated CVAs (cvCVAs) were also produced in order to check for spurious group separation.

The authors mention the discrete traits. I wonder why the authors did not include and score them in this paper, given that the first author has already published a paper regarding the lower molar and shows that there are taxon-specific patterns, although we must be cautious in comparisons of accessory cusps for taxonomic interpretations.

The inclusion of some discrete traits in OES and EDJ will be very interesting and could be use later by some researchers for phylogenetic analyses.

As the reviewer mentions, a paper discussing lower molar accessory cusps (including in H. habilis) has already been published. One of the major conclusions of this paper is that discrete traits should be used cautiously, as they frequently show a high level of variation at the EDJ, and the developmental underpinnings of these traits are poorly understood. Hence, we believe a separate investigation dedicated to discrete traits would be more appropriate. This would allow for a thorough exploration of their developmental complexities, something that exceeds the purview of our current manuscript.

Specific comments

Abstract

L5 and L9. Add east African before H. habilis... if the south african specimens are not included in the next version of the manuscript.

In our opinion it is most appropriate to attribute specimens in South Africa to Homo sp. (rather than H. habilis); however, we have now included SK 27 and SK 847 in our 'Homo sp.' sample. Their taxonomic affinities are assessed using CVA based typicality probabilities (Supplementary Table S7), and discussed in lines 234-236 and 515-520.

L 35. More references should be added from West Turkana (WT 42718) and from the Omo valley.

We have added these references.

L41. The reference of Leakey et al., 2001 should also be included as they state that “The transfer of this species (i.e. rudolfensis) to Australopithecus has been recommended, but Kenyanthropus may be a more appropriate genus”.

We have added this reference.

L41. It should also be mentioned that some studies (based on the same type of analysis (phylogenetic analyses), do not reach the same conclusions (Strait et Grine, JHE 2004; Dembo et al., JHE 2016; Caparros and Prat, IScience 2021). This comment and references should be added in the manuscript.

We have added a sentence to demonstrate this, and included these references.

L42. What is the proportion of dental remains? by mentioning “the large portion”.

We have reworded this sentence.

L76. Since specimens from southern Africa attributed to early Homo are not included in this study (see more general remark), “specimens of *H. habilis*” should be replaced by “some *H. habilis* specimens from eastern Africa”.

See above

L78. If one of the goals of this paper is to assess the homogeneity of *H. habilis* hypodigm, the authors should mention in the SI which hypodigm they considered (i.e. specify which specimens are included in *H. habilis*). Since this study is only conducted in one part of the hypodigm, due to the study of dental remains and only some from Olduvai and Kooki Fora, this sentence is bit exaggerated and goes much further than what can be argued in this paper.

*We have revised this statement to clarify that our focus is on the dental remains and not the entire hypodigm. Additionally we have added a supplementary table detailing our view of the *H. habilis* dental hypodigm (Table S3). The majority of specimens securely attributable to the species that preserve whole tooth crowns were included in this analysis, and we believe the sample is sufficient to assess the homogeneity of the species from a dental perspective.*

L83. Method. Why the authors did not used bg PCA (see general remark)?

See our detailed response above.

L86. It should be important in all the analyses to distinguish *A. africanus* and *A. afarensis* (two different ellipses), as these species originate probably from two different regions with possibly different evolutionary histories.

*Given the high density of information required to be incorporated into the relatively compact PCA plots, our aim is to maintain their simplicity for readability. Therefore, we chose to represent the data using broad *Australopithecus* and later *Homo* groups. However, we have produced additional interactive html versions of each PCA plot in which each specimen is individually identifiable, which are included as supplementary data files.*

For the purposes of figures, analyses and interpretations, it would be good to plot (and name) all *Homo erectus* specimens (which are in fact african *Homo erectus*, as there is no asian *Homo erectus* in the sample), instead of putting *H. erectus* with a line (Figure S6, Figure 3..).

*We have now added specimens from Sangiran to the *H. erectus* sample. The individuals can be identified in the interactive html PCA plots included as supplementary data in the revised submission.*

In Figure 3. There seem to be two specimens close to OH 16, which are very different from the other specimens). What are the labels of these specimens?

*The two specimens close to OH 16 in the LP3 PCA are modern humans (ULAC 536 and 801) that have a reduced dentine body heights relative to other *H. sapiens* in the sample (although there are other features that distinguish them from OH 16).*

L 93. Figure S5 illustrates the term used to describe crown height at the EDJ, but not why the

increase of the height of the dentine body is the notable shape between Australopithecus and later Homo. Figures and analyses that support this conclusion should be included here.

We have edited the text here to be clear that Figure S5 provides an explanation of terminology (this is also referenced in the methods section). Wireframe images of PC extremes illustrate that an increase in dentine body height is important in distinguishing later Homo from Australopithecus, which are shown in Figs. 3b, 4b, 5b, Supplementary Figs. S8 and S9.

Table 2. It would be interesting to include in this table clearly the shape characteristics for the african Homo erectus specimens, and for Australopithecus afarensis and Australopithecus africanus. L 102. It would be interesting to include in the supplementary informations, the condition (cf. Table 2) of all the Homo habilis studied here (including the Kooki fora specimens).

We have added two supplementary tables, one that outlines the condition seen in H. habilis specimens from Koobi Fora (KNM-ER 1802 and 1813) and A.L. 666-1, and one that outlines the condition in African H. erectus specimens (KNM-ER 806, 992, 1507 and OH 22). We have also added a column to the first of these tables that outlines any differences between A. africanus and A. afarensis in these features. We have also made corrections and updates to Table 2 in order to ensure that all character state assessments are consistent across this broader range of comparisons (including H. erectus specimens), and that the character state assessments hold true in comparison to the mean shapes of A. africanus and A. afarensis (as opposed to only the overall Australopithecus mean, previously). We have also updated the main text accordingly.

L104. How many specimens of Homo habilis retain the primitive conditions?

This depends on the feature being considered – the details are outlined in table 2 and discussed in the remainder of this paragraph (Line 121 onwards)

L116. Could you give the number of specimens and their accession number, instead of “several”. Homo erectus should be changed by african Homo erectus, as there is no asian Homo erectus in the sample.

This section has moved (now line 202 onwards) and has been expanded to more thoroughly discuss the positions of individual H. erectus specimens in PCAs. As discussed above, specimens from Sangiran have been added, so our H. erectus sample now includes specimens from Africa and Asia.

L 119. Since some african Homo erectus specimens are mentioned in the text to be distinguished from Australopithecus, their label (name) must appear clearly in all PCA Figures.

As discussed above, we have made html versions of all PCAs in which each specimen can be individually identified. There is not enough room to do this in the main plots.

L152. I am not sure if this sentence refers to Supplementary Fig.S3?

This has been corrected to Supplementary Fig. S6 (Line 170)

L 175. In Figure 6, it would be good to make the distinction between A. africanus and A. afarensis and to label in this figure AL 199-1 and AL 200-1a, as they are mentioned in the text.

As in the PCAs, these figures have a tendency to be very busy, so we have produced html versions in which each specimen can be individually identified.

L180. I am not sure if this sentence refers to Supplementary Table S3?

This refers to permutation tests of group differences for shape and size, which has now moved to Supplementary Table S6 – this has been updated (line 266)

L199. I am not sure if this sentence refers to Supplementary Table S3?

This refers to permutation tests of group differences for shape and size, which has now moved to Supplementary Table S6 – this has been updated (line 285)

L 202. Figure S3. It might be also interesting to make the same figures with H. erectus vs. Australopithecus and H. erectus vs. Later Homo. Since the centroid size across the tooth row of Homo erectus is close to that of H. habilis, one would expect H. erectus to be included in the range of variation of Australopithecus. The results of these two graphs concerning H. erectus should be therefore discussed.

We updated the main text Figure 6 to compare H. habilis to Australopithecus, H. erectus and later Homo, and updated Supplementary Fig S17 to compare H. erectus to Australopithecus, H. habilis (plus Homo sp.) and later Homo, so all combinations including H. erectus are now shown. Additionally, adding specimens from Sangiran to the H. erectus sample allows for H. erectus to be included in permutation tests of group differences (previously the samples were too small). These results are presented in Supplementary Table S6. The H. erectus size results have been added to the results section (line 287-292) and discussion (line 407-414).

L203. As KNM-ER 992, 1507 and KNM-WT 15000 are mentioned in the text, their labels should be seen in Supplementary Figure S13.

A number of H. erectus specimens are now mentioned in the text, such that it is not practical to put individual labels on each in the centroid size figures – as for the PCAs, we have instead made html versions of the plots in which each specimen is individually identifiable (available as Supplementary Data)

DISCUSSION

It might also be interesting to discuss whether or not african Homo erectus specimens also retain some Australopithecus conditions.

We have added discussion of this (line 401 onwards).

L218. Since it is only a part of the Homo habilis hypodigm that has been studied here (e.g. no south African specimen, no specimens from the Omo valley and only dental material), it should be more appropriate to be more specific/less general in purpose. Write differences between the Homo habilis specimens studied here (mainly from Olduvai and Koobi Fora) instead of differences between Homo habilis..., unless, in the revised version, other specimens, especially those from South Africa, are included.

We have edited this line to read “we fail to find statistically significant differences between our H. habilis and Australopithecus samples”, and as discussed above we have added specimens from Omo and South Africa to the sample (although they are separated from the main H. habilis sample).

L221. Referring to Table 1, add that the sample size of Upper M3 in this study for Homo habilis is n=3.

There are 4 H. habilis UM3s, but we have added this information to this section (which now discusses UM2s and UM3s)

In the discussion, the results presented here should be placed in a broader perspective, including results based on phylogenetic analyses that include some Homo habilis and Australopithecus specimens using craniodental traits (Strait et al., JHE 2004; Dembo et al., JHE 2016; Caparros and Prat, IScience 2021) or dental traits (Irish et al., Science, 2013).

We have added expanded the discussion to place the results in a broader perspective, with reference to the phylogenetic studies mentioned (line 383-400).

L238. To make this sentence more precise, could the authors list the dental traits, and if necessary discuss them with the literature.

The OES dental traits we are referring to are discussed in the paragraph following this sentence, with references to the literature.

L256. The authors mention the discrete traits. I wonder why the authors did not include and score them in this paper, given that the first author has already published a paper regarding the lower molar and shows that there are taxon-specific patterns, although we must be cautious in the comparisons of accessory cusps for taxonomic interpretations.

The inclusion of some discrete traits in OES and EDJ will be very interesting and could be use later by some researchers for phylogenetic analyses.

Furthermore, this list of discrete traits and results could be discussed with the results from Irish et al., where habilis/rudolfensis are related to H. erectus and not to A. africanus.

See explanation above for why we did not include discrete traits

L 264. Reference 48 (Leakey et al., 1974) should probably be deleted, as the number of specimens was very different at that time and as mentioned by Leakey et al., 1974 themselves, “A second genus would incorporate many of the gracile specimens from Sterkfontein presently referred to A. africanus, perhaps certain specimens from East Rudolf including KNM-ER 1813, and possibly some from Olduvai such as OH 24 [...] These remarks are necessarily speculative. A more detailed review of hominid systematics is being prepared in collaboration with B. A. Wood.. “ and B. Wood did not include these specimens in Australopithecus in the monograph concerning Koobi Fora in 1991.

We have deleted this reference

L280. Add east african before H. habilis, as the sample studied here is not the hypodigm of H. habilis. In this part of the discussion, a paragraph about east African H. erectus should be added, and especially when OH 16 and OH 13 are mentioned.

We have addressed the comment about the H. habilis hypodigm above. We have also expanded the discussion of H. erectus specimens (line 401 onwards)

Conclusion

L 355. "suggesting that dental changes that are associated with later species of the genus Homo". Is it also true for Homo erectus, or does this sentence refer only to the later Homo sample?

This sentence refers only to the later Homo sample used here. To make this clearer, we have changed the wording from "later species of the genus Homo" to "later Homo" so that this sentence is consistent with the wording used in the rest of the manuscript.

L359-L 363. This part should probably be discussed in the discussion paragraph and placed in a more general perspective including results based on phylogenetic analyses that include Homo habilis and Australopithecus specimens (Strait et al., JHE 2004; Dembo et al., JHE 2016; Caparros and Prat, IScience 2021, Irish et al., Science 2013) (see comments on the discussion).

We have added expanded the discussion to place the results in a broader perspective, with reference to the phylogenetic studies mentioned (line 383-400)

L364. The authors mention that their results highlight excessive levels of dental variation of the H. habilis hypodigm, but this study concerns only one part of the hypodigm. It would be nice if the authors would mention in the supplementary information, what they consider to be the Homo habilis hypodigm (see also my previous comment in general remark).

See above comments on the H. habilis hypodigm, and Table S3.

Supplementary informations.

Supplementary note 1.

I am not sure if autapomorphy is the most appropriate term. How do you test it? How are you sure that the traits mentioned are autapomorphies (taxon specific) and not synapomorphies or symplesiomorphies?

We have changed the title to 'H. habilis derived dental traits'

Figure S9, S10 and Figure 5. What are the distributions for H. erectus, Neandertals, Modern humans, A. africanus and A. afarensis.

Modern humans are already included in the distribution figures for Procrustes distance (now Figure S10) and dentine body height (Figure S11), as well as those in summary figures within the main text (Figures 3, 4, 5). For fossil groups, we found that the sample sizes are not large enough to give reliable or informative distributions, so we chose to only include extant taxa for which there are sufficient sample sizes.

Supplementary tables.

Table S1 and Table S2. It is worth mentioning where the data come from. From this study? If not (which is probably the case, as there are sometimes more specimens in Table S1 than in Table 1), could the authors add the reference.

This data comes from published measurements; the references are listed in Supplementary Note 1. We have added a note to the tables to refer the reader here for more information.

Reviewer #2

This study conducts a large-scale analysis of enamel-dentine junction morphology in *Homo habilis*, with a large comparative sample of fossil hominins and extant apes. The paper is well-written with refined figures and clear language throughout. Their analyses focus on variation in EDJ morphology in *Homo habilis* using micro-CT scans and Procrustes analyses. All methods employed are appropriate. Using EDJ variation across several different teeth, the authors find that OH 16 is more similar to later *Homo*, while most other specimens attributed to *Homo habilis*, including OH 7, the holotype, look much more like *Australopithecus*. Recent publications investigating dental morphology in hominins have also found evidence that the dental morphology of early *Homo* looks very *Australopithecus*-like (e.g., Brasil et al. 2020; Jiménez-Arenas 2020).

Overall, this is a very strong study with interesting findings that are relevant to human evolutionary studies and paleoanthropology. I think the paper is well written and well-presented, and the results support the interpretation. However, I have three main concerns with the manuscript which I feel require addressing.

1) First and most importantly, I have serious concerns about the data availability for this project. As currently presented, readers are unable to assess data reliability. Additionally, the study does not offer replicability, a key requirement of all science. As written, the data availability statement does not meet the requirements of the journal. If scan data cannot be made directly available due to concerns by owning nations or repositories, these conditions should be discussed directly in the data availability statement. Under what conditions can scans be accessed? How did the authors access these scans? Additionally, the exact requirements required for access to landmarked datasets should be included in the data availability statement. Far too many researchers have reported being denied access to data despite data availability statements claiming access will be granted “upon reasonable request”. Please indicate exactly what a reasonable request is, so that future researchers wishing to replicate and build upon this research have transparency about the location and availability of these data. The goal of science is to build up, not to hold up. If these data are ready for publication, then they are ready to be made available for scientific purposes. Please address all journal requirements and revise your data availability and/or statement.

JOURNAL REQUIREMENTS

An inherent principle of publication is that others should be able to replicate and build upon the authors' published claims. A condition of publication in a Nature Portfolio journal is that authors are required to make materials, data, code, and associated protocols promptly available to readers without undue qualifications.

Data availability subject to controlled access: the data availability statement should include the following information: reasons for controlled access (eg., privacy, ethical/legal issues), conditions of access must be described precisely including contact details for access requests, timeframe for response to requests, restrictions imposed on data use via data use agreements. A copy or link to the data use agreement should be provided if requested by editors. Restrictions on controlled access datasets including restrictions on downstream data reuse or authorship requirements must be clearly described in manuscript and to editors at the time of submission. Editors may decline further consideration of the manuscript after evaluation if restrictions are found to be unduly prohibitive.

Third party data: when data obtained from third parties cannot be made available, the restrictions should be clearly stated in the data availability statement. Authors must make data available for purposes of peer review, if requested by reviewers, within the terms of a data use agreement and if compliant with ethical and legal requirements.

We have made several changes in response to this comment and the journal's data availability requirements. Firstly, we have made geometric morphometric landmark data available through the publications section of The Human Fossil Record (<https://human-fossil-record.org>). Currently this data can be accessed using a reviewer login (Username: Davies_Habilis Password: X2UnnJWsbCeA), but will be made publicly available upon publication. GM code is also available from the same repository, which allows readers to replicate our results.

We have also edited the data availability statement to be clear about the available data, and the restrictions that exist regarding access to museum collections and associated CT-scans.

2) A survey of the literature cited indicates that the authors have not cited anything that has been released in the last 5 years, unless it was written by them. This is an oversight of the landscape of contemporary paleoanthropology. As such, the authors have missed key citations from scientists active in the field, many of which play directly into the authors' arguments and analyses. Please see bullet list of a few examples below. I recommend citing these relevant, modern publications in your revisions:

Bruner and Beudet (2023) – discussion on endocranial variation in *H. habilis*
<https://doi.org/10.1016/j.jhevol.2022.103281>

Brasil et al. (2020) – dental data place *H. habilis* and other early *Homo* as more similar to *Australopithecus*
<https://doi.org/10.1007/s00114-020-01696-9>

Grine et al. (2019) – KNM-ER 64060, affiliation with *H. habilis*
- <https://doi.org/10.1016/j.jhevol.2019.03.017>

Monson et al. (2020) – evolution of EDJ in humans. Specifically discusses dentine body height variation
<https://doi.org/10.1073/pnas.2008037117>

Jiménez-Arenas (2020) – dental research on variability in *H. habilis* with suggestion that it exceeds one species.

<https://doi.org/10.7203/sjp.27.1.18090>

Pan et al. (2020) – EDJ variation and taxonomy in fossil apes
DOI: 10.16359/j.cnki.cn11-1963/q.2020.0023

Another good citation to include is this review of early Homo taxonomic discussions by Antón (2012):

Antón (2012) – taxonomic instability of *H. habilis*
<https://doi.org/10.1086/667695>

We have integrated the suggested references, as well as a number of other relevant recent references.

3) In the methods, the authors mention that they reconstructed several specimens prior to analysis (lines 433-435). Since we are not able to see all specimens, please indicate in table or text all specimens for which reconstructions were performed.

We have added a column to the sample table (Supplementary Table S8) that lists for each specimen whether cusps were reconstructed, and we have also added a Supplementary Figure (S6) that shows the cusp reconstructions for *Homo habilis* specimens.

Reviewer #3

This paper presents an analysis of the enamel dentin junction of a large sample of hominin teeth to assess the morphology of the EDJ in *Homo habilis*. I found this manuscript to be very well written and methodologically sound. There were certain elements of this paper that I particularly appreciated, such as the inclusion of Table 2 which provides a nice summary of morphological differences between the various hominin groups rather than leaving the reader to interpret undefined shape space on their own.

Despite these strengths, however, I do think one key element is conspicuously missing. One of the stated goals of this paper was to evaluate the homogeneity and affinities of the *H. habilis* hypodigm. Given that *H. habilis* has been reconstructed as sharing its closest phylogenetic affinities to *Homo rudolfensis*, as well as the ongoing controversy over the *H. habilis* vs. *H. rudolfensis* hypodigms, this taxon represents an obvious comparative sample for the questions at hand—arguably much more so than the very large Neanderthal sample which has been included in the study. Lines 331-333 note that several *H. rudolfensis* specimens (OH 65, KNM-ER 60000, KNM-ER 62000, and KNM-ER 62003) could not be included in the study, but give no further explanation. I can probably read between the lines vis a vis including OH 65, but am confused as to why KNM-ER 60000, KNM-ER 62000, and KNM-ER 62003 could not be included given that one of the key authors who originally described these specimens is also an author on the present study.

Micro-CT scans of Kenyan fossils in our sample were made in Nairobi when a portable CT scanner was temporarily imported at major costs. However, this happened before KNM-ER 60000, KNM-ER 62000, and KNM-ER 62003 were found. Exporting these fossils to micro-CT scanners in other countries is complex not realistic in the context of this study (especially as the installation of a

micro CT at TBI in Nairobi was anticipated, although disrupted by the Covid period).

On a related note, I see that there is a very large *A. africanus* comparative sample and wonder whether the authors might consider including *A. sediba* as well. Given its purported links to *Homo* (Berger et al., 2010), and specifically to *H. habilis* (e.g., Dembo et al., 2015), this would be an interesting sample to address in the present study.

A study focusing on *A. sediba* is indeed in progress, but it is at a preliminary stage, and requires additional micro-CT scans that have been impossible to create due to a lack of functioning scanning facilities at the University of the Witwatersrand. Additionally, for most tooth positions there is only one suitably unworn specimen available, limiting the strength of any comparisons.

Another issue that needs to be addressed in the introduction of the paper is the definition of the *H. habilis* hypodigm. Lines 78-79 state that one of the goals of the paper is “to assess the homogeneity of the *H. habilis* hypodigm as currently defined.” Given that this is a controversial taxon, it would be useful to include references here (e.g., as currently defined by whom?) For example, several studies have included Stw 53 in their *H. habilis* hypodigm (Strait and Grine, 2004; and by extension: Dembo et al., 2015; 2016; Mongle et al., 2019; 2023). Was the hypodigm in this manuscript restricted to only include specimens from eastern Africa, do the authors disagree with the hypodigm used by other studies, or was Stw 53 excluded due to its high degree of dental wear? Likewise, Wood (1992) has included other specimens like KNM-ER 1805 in *H. habilis*, but that has not been included here. Was this specimen excluded because the KNM-ER 1805 dentition was considered too damaged to assess or do the authors disagree with Wood (1992)? I would presume that the authors are following the *H. habilis* hypodigm suggested by Spoor et al. (2015), but this should be explicitly stated.

In order to be clearer on this issue, we have created a Supplementary Table (S3) which outlines the hypodigm that we consider to belong to *H. habilis*. For some specimens, such as KNM-ER 1805, they were excluded as the CT scans are not able to provide distinction between enamel and dentine – we have outlined these cases in the table. The majority of specimens from South Africa are not securely attributed to species level, which was why they were initially excluded from the sample. However, we recognise the importance of expanding the geographical range of the sample, so we have added several specimens that have been previously attributed to *Homo habilis* by some authors. We limited this to specimens found by Zanolli et al. 2022 to likely belong to *Homo* on the basis of EDJ morphology so as to avoid repeating analyses – which excludes StW 53. As such we have added SK 27 and SK 847 to our sample.

While I felt that the majority of the study was methodologically sound, I do have some concerns related to the reconstruction of cusp tips for landmark placement in fossil specimens with moderate wear (e.g., OH 7, OH 13, OH 16, AL 666, ER 1813 M1 protoconids). The authors state that these missing portions of the mesh were reconstructed in Geomagic following a protocol established in the senior author’s dissertation. In my own experience, these reconstructions can vary wildly based on the preserved portions of the cusp and the specific Geomagic setting that was used for interpolation. Given that relative protoconid height was one of the characteristic features described for the *H. habilis* dentition (Table 2), it would be useful for readers to be able to evaluate figures of these reconstructed meshes in the supplemental material.

We have added a Supplementary Figure (S6) that shows the cusp reconstructions for each *Homo habilis* specimen so that readers can evaluate the reconstructions themselves. We have also added

a column to the sample table (Supplementary Table S8) that lists for each specimen whether cusps were reconstructed.

On lines 353-356, the authors conclude: “We find that the postcanine EDJ morphology of a number of key specimens of *H. habilis*, including the type specimen OH 7, is very similar to that of *Australopithecus*, suggesting that dental changes that are associated with later species of the *Homo* genus, were not present in the earliest members of our genus.”

While this conclusion appears to be supported by the one PCA provided in the main text (Fig. 5a), the picture becomes a bit more nuanced when considering results across the full dataset in the supplementary information. For example, the lower M1-M3 PCAs (Fig S6) also show *H. erectus* completely overlapping with *Australopithecines* in morphospace. Had *H. rudolfensis* been included, I would imagine it would find itself overlapping those polygons as well. All of this to say, I think your results speak to a clear difference between Humans/Neanderthals and earlier members of the genus *Homo*, but it is an overstatement of the results to claim that these results indicate “future research should address the implications of this for the genus level designation of the *H. habilis* hypodigm” (lines 362-363)

It is true that some *H. erectus* specimens that overlap with *Australopithecus* in PCAs, however this varies by tooth position and specimen, and overall we find that the *H. erectus* specimens are more derived than those of *H. habilis*. We have taken some steps in the revisions to demonstrate this more clearly, including the addition of CVAs that allow for better distinction between *Australopithecus*, *H. habilis* and *H. erectus*, as well as the inclusion of *H. erectus* in permutation tests. These analyses demonstrate that there are significant differences between *H. erectus* and *Australopithecus* in EDJ shape in all mandibular tooth positions, as well as maxillary P^4 and M^2 , and that there are shape differences that characterise the *H. erectus* sample and differentiate them from *Australopithecus* (outlined in lines 223-229 and 401-414). We have also created a table that outlines the expression of key EDJ features in *H. erectus* specimens to match Table 2 (Supplementary Table S5), and we have expanded our discussion of individual *H. erectus* specimens (line 205-223 and 415-431). This should demonstrate more clearly the variation that is present within the *H. erectus* sample, and show that, for example, KNM-ER 992 is clearly derived in a number of features. On the other hand, we find that one specimen, KNM-ER 1507, is particularly primitive in several aspects, which is evident in new CVA analyses. Overall, the specimen accords more closely with *H. habilis* or *Australopithecus* in EDJ shape (P_3 - M_2) and we discuss the taxonomic affinities of the specimen in the light of these results.

Overall, our results demonstrate the differences between *H. erectus* and *H. habilis*, with the former being more derived in several key aspects, while the *H. habilis* largely retains an *Australopithecus*-like EDJ morphology (with the exception of a small number of features, which are discussed).

We have also edited the line about the “implications for genus level designation of *H. habilis*” to be clear that we are referring to an already ongoing discussion, for which we have added references, regarding the morphology of the skeleton as a whole (line 533-536)

Table S4: need to indicate what * next to species designation means in table key (e.g., OH 22, OH 23)

We have updated the footnotes of the table to explain the asterisk, which refers to the fact that these specimens are separated from the main *H. erectus* sample in figures due to the younger age of these specimens

Reviewers' Comments:

Reviewer #1:

Remarks to the Author:

I would like to thank the authors for their responses to the comments and suggestions made, with the expansion of the sample of early Homo and Homo erectus specimens, the GM analyses and the updating of the figures and tables.

I see no problem with publishing this article.

I just have a few specific comment.

Abstract

L10-L11. For the sentence "Supporting the hypotheses that H. habilis hypodigm has more in common with Australopithecus than Homo", it would probably be more appropriate to add "dental "before hypodigm (to be on the same line as your answer see below and L91) and to put later Homo (since H. habilis belongs to the genus Homo (in this sentence)), and to be consistent with the text of the manuscript (e.g. L87, L309, L399, L528)

->Supporting the hypotheses that H. habilis dental hypodigm has more in common with Australopithecus than later Homo"

Responses for previous comment L 78.

We have revised this statement to clarify that our focus is on the dental remains and not the entire hypodigm. Additionally we have added a supplementary table detailing our view of the H. habilis dental hypodigm (Table S3). The majority of specimens securely attributable to the species that preserve whole tooth crowns were included in this analysis, and we believe the sample is sufficient to assess the homogeneity of the species from a dental perspective.

L74. relvence->relevance

L314. Delete one "."

More broadly..-> more broadly.

L433. Add dental

-> H. habilis dental hypodigm, cf previous fist remark

L436, L 438, "ref 66" should be written 66

L443, "ref 68" should be written 68

L460 hominines, hominins or hominids? For extant hominines

L501. Add a space before Ma

2.3Ma->2.3 Ma

L 579.Add dental

-> this early dental Homo hypodigm, cf previous fist remark

Table S3. Add dental in the title of Table S3

-> dental Homo hypodigm, cf previous fist remark

Table S3. Why "?" for in sample ?

Reviewer #2:

Remarks to the Author:

This is a strong paper on variation in enamel-dentine morphology in fossil hominids. The authors have addressed most of my concerns, and I thank them for now including the data as part of the manuscript. The revised instructions on how to access scan data are also appreciated (note that it should be "scan data used in this study are", and "these data were used").

I have one other major concern before the manuscript is suitable for publication.

The authors state that "A number of studies have demonstrated that the morphology of the EDJ is

taxonomically informative". In fact, the entire study rests upon this supposition. However, the 6 references cited for this phrase are focused overwhelmingly on fossil dentition and hominid taxonomy. To justify this statement, please include a reference that is focused on EDJ variation in extant groups.

Without a strong study examining EDJ variation in extant primates (preferably not just apes), it is very difficult to state that EDJ varies taxonomically. The taxonomy of fossil hominids is convoluted and much under debate. And it cannot be reaffirmed using a trait for which the variation has only been investigated in other fossils - this is circular logic on its ability to differentiate taxonomic groups.

I feel strongly that a description of how EDJ varies in extant animals is essential for the argument made in this paper. To this end, why isn't EDJ morphology for the extant apes shown in the multivariate space (PC) figures? It is important to see how the range of variation for these extant apes compares to the fossil taxa.

Minor comment: Look out for species italics in references (e.g., 40. White et al.)

Line 383 - hypodigms

Line 434 - single "fossil" species - Please clarify throughout when your assessment of EDJ variation applies only to fossils or more generally across mammalian biology.

Study sample n= has changed - change throughout for consistency

Reviewer #3:

Remarks to the Author:

I would like thank the authors for their careful consideration of reviewer comments. I am satisfied with the revised version, although I would caution that the newly added text could use another close review for typos throughout (e.g., lines 386 and 394 in the track-changes version of the document).

We would like to thank the reviewers for their comments. We have made a number of changes in response, as outlined in detail below. Additionally, we have corrected a minor mistake in centroid size figures (Fig. 6 and S13), which do not affect the results, and we have corrected some spelling mistakes in Supplementary Table S8.

Reviewer #1 (Remarks to the Author):

I would like to thank the authors for their responses to the comments and suggestions made, with the expansion of the sample of early Homo and Homo erectus specimens, the GM analyses and the updating of the figures and tables.

I see no problem with publishing this article.

I just have a few specific comment.

Abstract

L10-L11. For the sentence “Supporting the hypotheses that H. habilis hypodigm has more in common with Australopithecus than Homo”, it would probably be more appropriate to add “dental “before hypodigm (to be on the same line as your answer see below and L91) and to put later Homo (since H. habilis belongs to the genus Homo (in this sentence)), and to be consistent with the text of the manuscript (e.g. L87, L309, L399, L528)

->Supporting the hypotheses that H. habilis dental hypodigm has more in common with Australopithecus than later Homo”

Responses for previous comment L 78.

We have revised this statement to clarify that our focus is on the dental remains and not the entire hypodigm. Additionally we have added a supplementary table detailing our view of the H. habilis dental hypodigm (Table S3). The majority of specimens securely attributable to the species that preserve whole tooth crowns were included in this analysis, and we believe the sample is sufficient to assess the homogeneity of the species from a dental perspective.

In the abstract (line 10-11) we have changed ‘Homo’ to ‘later Homo’ for clarity. We would prefer to keep ‘hypodigm’ rather than ‘dental hypodigm’ in this instance because we are referring to the wider context of our results here (as in parts of the discussion and the conclusion). However, we have been through the text to ensure that ‘dental hypodigm’ is used whenever we are referring specifically to teeth (lines 406 and 545).

L74. relvence->relevance

L314. Delete one “.”

More broadly..-> more broadly.

L433. Add dental

-> H. habilis dental hypodigm, cf previous fist remark

L436, L 438, “ref 66” should be written 66

L443, “ref 68” should be written 68

We have made all of the above changes

L460 hominines, hominins or hominids? For extant hominines

In this case, we are referring to the extant Pan, Gorilla and H. sapiens sample included in Figure S11, so extant hominines is correct. We have added a reference to this figure here to make this clearer.

L501. Add a space before Ma

2.3Ma->2.3 Ma

L 579.Add dental

-> this early dental Homo hypodigm, cf previous fist remark

We have made all of the above changes

Table S3. Add dental in the title of Table S3

-> dental Homo hypodigm, cf previous fist remark

We have changed the title of Table S3 to read “Homo habilis dental hypodigm – specimens preserving whole tooth crowns”

Table S3. Why “?” for in sample ?

This column lists whether each specimen is included in the sample of this study. For clarity, we have changed this to ‘Included in sample?’ and we have added to the header a more detailed explanation of the information contained with the table.

Reviewer #2 (Remarks to the Author):

This is a strong paper on variation in enamel-dentine morphology in fossil hominids. The authors have addressed most of my concerns, and I thank them for now including the data as part of the manuscript. The revised instructions on how to access scan data are also appreciated (note that it should be "scan data used in this study are", and "these data were used").

We have made these changes

I have one other major concern before the manuscript is suitable for publication.

The authors state that "A number of studies have demonstrated that the morphology of the EDJ is taxonomically informative". In fact, the entire study rests upon this supposition. However, the 6 references cited for this phrase are focused overwhelmingly on fossil dentition and hominid taxonomy. To justify this statement, please include a reference that is focused on EDJ variation in extant groups.

Without a strong study examining EDJ variation in extant primates (preferably not just apes), it is very difficult to state that EDJ varies taxonomically. The taxonomy of fossil hominids is convoluted and much under debate. And it cannot be reaffirmed using a trait for which the variation has only been investigated in other fossils - this is circular logic on its ability to differentiate taxonomic groups.

I feel strongly that a description of how EDJ varies in extant animals is essential for the argument made in this paper.

The reviewer raises an important concern that we would like to address in some detail. The morphology of the EDJ has indeed been found to vary taxonomically in extant groups. For example,

Olejniczak et al. (2004, 2007) found that the configuration of the maxillary molar EDJ differs between primate species in 2D sections, and that it is possible to classify primate taxa based on EDJ morphology. Using 3D geometric morphometrics, Skinner et al. (2009) were able to discriminate between species and even subspecies of Pan using mandibular molar EDJ shape, and Davies et al. (2019) correctly classified extant ape mandibular premolars based on EDJ morphology (within a broader extant and fossil hominoid sample).

*It is also important to note that studies of EDJ morphology in fossil hominins have produced results that are consistent with well-established hypodigms that are based on much broader criteria (i.e. not just EDJ shape or other aspects of tooth morphology). For example, Martin et al. (2017) (<https://doi.org/10.1016/j.jhevol.2016.12.004>) demonstrated clear differences in EDJ morphology between modern humans and Neanderthals. These differences in EDJ morphology are consistent with morphological evidence across the skeleton of *H. sapiens* and Neanderthals, as well as molecular data confirming the distinction between the two groups. In this case, the ability of the EDJ to discriminate between taxa does not rely on circular logic – we are finding a clear taxonomic signal consistent with other independent lines of evidence.*

In order to demonstrate more clearly the utility of EDJ morphology for systematics in extant groups, we have reworded the sentence in the introduction (line 72-74) to read:

“A number of studies have demonstrated that the morphology of the EDJ is taxonomically informative in extant primates^{42,44–46} and fossil hominoids^{43,47–51}”

This includes the addition of the following references:

Olejniczak et al. (2004): [https://doi.org/10.1016/S0940-9602\(04\)80087-6](https://doi.org/10.1016/S0940-9602(04)80087-6)

Olejniczak et al. (2007): <https://doi.org/10.1016/j.jhevol.2007.04.006>

Skinner et al. (2009): <https://doi.org/10.1002/ajpa.21057>

Davies et al. (2019): <https://doi.org/10.1016/j.jhevol.2019.06.004>

To this end, why isn't EDJ morphology for the extant apes shown in the multivariate space (PC) figures? It is important to see how the range of variation for these extant apes compares to the fossil taxa.

*The morphology of the EDJ in extant apes is quite different to that of hominins, and as such including the apes in PCAs would lead to the first principal component(s) being dominated by the aspects of EDJ shape that distinguish between apes and hominins. This would obscure the components of EDJ shape that differ among hominins, so it is important to exclude the apes here in order to effectively explore the EDJ shape differences within the hominin sample. However, the supplementary information does contain two figures (Figs S9 and S10, and main text figures 3, 4, 5) that quantify the within- and between group variation in extant apes and *H. sapiens*, both in overall EDJ shape (through Procrustes distances) and in dentine body height. This provides a context within which the variation within early *Homo* can be understood.*

Minor comment: Look out for species italics in references (e.g., 40. White et al.)

We have fixed these errors in the reference list

Line 383 – hypodigms

We have corrected this

Line 434 - single "fossil" species - Please clarify throughout when your assessment of EDJ variation applies only to fossils or more generally across mammalian biology.

In this case, we are referring to our results comparing the variation in *H. habilis* to that observed in extant species (*H. sapiens*, *G. gorilla*, *P. troglodytes*). This is outlined in the results section in more detail, and the results are included in the figures referenced in this line. For clarity, here and elsewhere we have added references to the supplementary figures summarizing these results

Study sample n= has changed - change throughout for consistency

We have updated this in the abstract, and it was already updated in the 'Study sample' section.

Reviewer #3 (Remarks to the Author):

I would like thank the authors for their careful consideration of reviewer comments. I am satisfied with the revised version, although I would caution that the newly added text could use another close review for typos throughout (e.g., lines 386 and 394 in the track-changes version of the document).

We have reviewed the newly added text and corrected for typos.

Reviewers' Comments:

Reviewer #2:

Remarks to the Author:

I am satisfied with the revisions.